



# Modeling of GPS total electron content over the African low latitude region using empirical orthogonal functions

Geoffrey Andima[a,*], Emirant B Amabayo[a,b], Edward Jurua[a], Pierre J Cilliers[c]

[a]*Department of Physics, Mbarara University of Science and Technology, Mbarara, Uganda*
[b]*Department of Physics, Busitema University, Tororo, Uganda*
[c]*South African National Space Agency (SANSA) Space Science, Hermanus, South Africa*

## Abstract

In this paper, an empirical total electron content (TEC) model and trends in TEC over the African low latitude region are presented. GPS-derived TEC data from Malindi, Kenya (geographic coordinates 40.194°E, 2.996°S) and global ionospheric maps (GIMs) were used. We employed empirical orthogonal function (EOF) analysis method together with least square regression to model the TEC. The EOF-based TEC model was validated through comparisons with GIMs, GPS-derived TEC and TEC derived from the International Reference Ionosphere-2016 (IRI-2016) model for selected quiet and storm conditions. The single station EOF-based TEC model over Malindi satisfactory reproduced the known diurnal, semiannual and annual variations in the TEC. Comparison of the EOF-based TEC model results with TEC derived from IRI-2016 model showed that the EOF-based model predicted the TEC over Malindi with less errors than the IRI-2016. For the selected storms, the EOF-based TEC model simulated the storm time TEC response over Malindi better than the IRI-2016. In the case of the regional model, the EOF-based TEC model was able to reproduce the TEC characteristics in the equatorial ionization anomaly region. The EOF-based TEC model was then used as a background in estimating TEC trends. A latitudinal dependence in the trends was observed over the African low latitude region.

*Key words:* TEC model, Empirical orthogonal functions, TEC Trends

## 1. Introduction

The features of the low latitude ionosphere are quite unique. During daytime, a double peaked ionization structure appears over the low latitude region, a phenomenon often referred to as the equatorial ionization anomaly (EIA). The EIA is normally explained in terms of the plasma fountain theory (Martyn, 1947; Moffett, 1979). The daytime E region eastward electric field in combination with the nearly horizontal Earth's magnetic field generate a large vertically directed E×B drift force at the dip equator that raises the plasma to higher altitudes. The raised plasma diffuses away from the geomagnetic equator under gravity and pressure gradient forces along the equipotential magnetic field lines to form ionization peaks at dip latitudes $\sim \pm 15°$, and a trough that extends over the dip equator (Appleton, 1946). Prior to the electric field turning westwards at night, it is enhanced (prereversal enhancement (PRE)) resulting in plasma uplift into regions of low recombination. Associated with the electron density at the EIA enhancement is a density gradient instability of the Rayleigh Taylor (RT) type which creates a spectrum of plasma irregularities that fill the post sunset low latitude ionosphere (Kelley, 2009). The EIA and the PRE vary with location, solar activity, season, and even on daily basis. These variations make it difficult to predict the characteristics of the low latitude ionosphere.

*Corresponding author
Email address: geoffrey.andima@gmail.com (Geoffrey Andima)
*Preprint submitted to Annales Geophysicae*                    *July 19, 2018*





The state of the ionosphere is of great importance in space based navigation systems such
as the Global Navigation Satellite Systems (GNSS). The total electron content (TEC) is of
particular interest to users of GNSS systems. For many practical purposes in the GPS, the
desired ionospheric parameter is the TEC. This is because many of the effects on transiono-
spheric satellite links (e.g time delay, Polarization, Faraday rotation, Doppler shift) are related
to TEC in one way or another (Kersley et al., 2004). The low latitude ionosphere exhibits
the highest values of TEC globally. Therefore, pronounced ionospheric effects are experienced
by radio signals transiting the low latitude ionosphere. Understanding the low latitude iono-
spheric dynamics in a bid to forecast its day to day conditions is key for advancement of space
technology and the improvement of GNSS accuracy.
Ionospheric variability over the low latitude region of Africa, based on TEC analysis, has
been reported before (e.g Adewale et al. 2011; Olwendo et al. 2012; Habarulema et al. 2013;
Andima et al. 2015). From these studies, the diurnal, seasonal, disturbed and quiet time TEC
characteristics over the region have been revealed. However, these analyses made use of TEC
data of either the same solar phase or the same solar cycle. With now a relatively longer
record of data in the achieves, it is imperative to extend these studies to the long-term TEC
characteristics over the African low latitude region for practical applications.
A common approach to TEC prediction is through modeling. Various TEC models (e.g
Anderson et al. 1987; Rawer and Bilitza 1990; Reinisch et al. 2004; Lean et al. 2011; Hajra
et al. 2016; Ercha et al. 2012; Chen et al. 2015; Ercha et al. 2012) have been developed;
however, many of these models are limited in geographical extend. A widely used model to
describe the global TEC climatology is the Intenational Reference Ionosphere (IRI) model
(Bilitza, 1990). The IRI is an empirical model synthesized from global data sets comprising of
ionosonde, radar and in situ measurements (Bilitza, 1990; Rawer and Bilitza, 1990). Averaging
and smoothing applied when deriving the model coefficients may limit its accuracy in capturing
peculiar features such as the TEC variability in the EIA region. Under such circumstances,
regional models are superior in characterizing the background TEC.
Long term trends in ionospheric parameters are indicative of the deviation in the ionospheric
parameters from their background values. Ionospheric trends are important in understanding
the changes in Earth's energy balance (Elias, 2011). Various studies (e.g Jarvis et al. 1998;
Bencze 2002, 2005; Lastovicka et al. 2006; Danilov and Mikhailov 1999; Bremer et al. 2012)
have reported on long-term trends in ionospheric parameters derived from ionosonde data.
A conclusion from these studies is that trends in the F2 layer critical frequency (foF2) and
F2 layer maximum electron density height (hmF2) are negative. Some studies on ionospheric
trends have also revealed latitudinal dependence of these trends (Danilov and Mikhailov, 1999).
Lean et al. (2011) using a database of global ionospheric maps (GIMs) reported that global
TEC trends are positive and are dependent on the geomagnetic latitude. Also cases of negligible
or no trends in TEC have been observed. For instance, results obtained by Lastovicka et al.
(2017) show a weak negative or no trend in ionospheric TEC. Despite the various studies on
trends of different ionospheric parameters, those relating to TEC remain limited, and hence
the question on the nature of ionospheric TEC trends still need to be answered. There is a
need to investigate whether TEC has negative (Lastovicka et al., 2017), or positive (Lean et al.,
2011), or no (Lastovicka, 2013) trend. The objective of this paper is therefore two fold: first
to attempt to model low latitude TEC and secondly to estimate trends in the variation of
the ionospheric TEC over the African low latitude region using actual TEC measurements by
means of regional GPS receivers, and data from the GIMs.

## 2. Data sets used

The International GNSS Service (IGS) operates a number of GPS ground based receivers over
the African low latitude region. In this study, data was obtained from one of the IGS receivers



located at Malindi, Kenya which archived data from 1995 to date (December 2017). Prior to 2008, the IGS receiver (station code MALI) was installed at 40.19439° E, 2.99591°S and then replaced with another (station code MAL2) installed at 40.19414°E, 2.99606°S. These receivers had nearly the same location and therefore sampled the same geographical region of the ionosphere. We obtained the Receiver Independent EXchange (RINEX) files from `ftp://cddis.gsfc.nasa.gov/` and then extracted the TEC along the line-of-site, slant TEC (sTEC), from the RINEX files using the GPS-TEC software of Boston College (Seemala and Valladares, 2011). This software uses the thin shell mapping function to map the sTEC to the vertical to obtain the vertical TEC (vTEC) at an assumed ionospheric height of 350 km. The vTEC for the different viewing geometries for satellites with elevation angles greater than 30° were averaged epoch-by-epoch to give a representation of the vTEC above the receiver. Due to data paucity from 1995 to 1998, only data from 1999 to 2017 were used in this study. Hourly averages of the daily TEC data were then calculated to minimize noise in the data. The hourly averages were organized into a data matrix $M_{d \times h}$ (day×hour) which was used to model and estimate trends in TEC over Malindi. To study the TEC over the African low latitude region, TEC from the GIMs, a reliable source of ionospheric data (Hernandez-Pajares et al., 2009), was used. Though these maps have been available since 1998, for comparison purposes with the GPS data, we have used data from 1999-2017. It is worthy noting that the two hourly GIMs were interpolated to hourly data.

## 3. Single station model over Malindi

### 3.1. EOF decomposition of the TEC data

Empirical orthogonal function (EOF) analysis is a well-known method that dates back to the work of Pearson (1901), and has been widely used in climate (Hannachi et al., 2007) and ionospheric (Dvinskikh, 1988; Zhang et al., 2009; Liu et al., 2008) data analysis. It involves reducing the dimensionality of the data by finding a set of few variables (EOF modes or basis vectors) that explain most of the variability in the data. This allows for the original data (TEC(d, h)) to be expressed as a linear combination of the few basis functions as

$$TEC(d, h) = \sum_{j=1}^{n} U_j(h) \times C_j(d),$$ (1)

where $C_j$ is the coefficient of the basis vector $U_j(h)$ with the index j running from 1 to n (the number of the retained EOF modes). We used the method of singular value decomposition (svd) to determine the EOF modes that explain most of the variability in the TEC data. The TEC data matrix M was decomposed into U and V, the left and right basis vectors respectively, and S a matrix of singular values of M according to the equation

$$M = USV^{\tau}$$ (2)

The first six EOF modes in matrix U and their corresponding coefficients obtained using equation 3 are shown in Figure 1. While Table 1 shows the percentage variability in the data explained by the different EOF modes.

$$C_j(d) = S \times V_j^{\tau}$$ (3)

Figure 1 (a) shows that the average diurnal TEC over MAL2 has a maximum value at about 11:00 UT and a secondary maximum at about 20:00 UT. The daytime peak at 11:00 UT is possibly due to increased ionization as the solar zenith angle is nearly zero over Malindi around this time. The secondary maximum could be due to an enhancement in the eastward




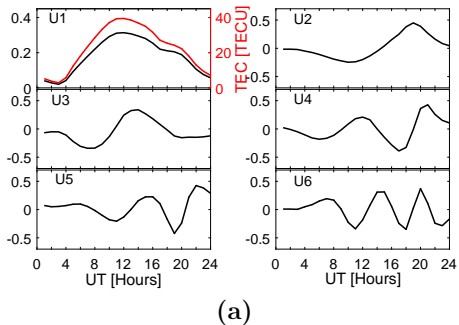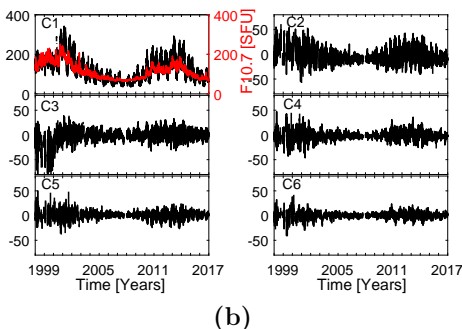

(a)           (b)

Figure 1: The first six basis functions (a) representing the diurnal variation and their coefficients (b) which show the long-term variation of TEC over MAL2

Table 1: Variance of the TEC data explained by the different EOF modes

| EOF mode | 1 | 2 | 3 | 4 | 5 | 6 |
|---|---|---|---|---|---|---|
| Explained Var. | 96.8 | 1.0 | 0.8 | 0.3 | 0.20 | 0.1 |
| Cumulative | 96.8 | 97.8 | 98.6 | 98.9 | 99.1 | 99.2 |

electric field before its westward reversal at night. Though the physical interpretation of the
basis functions are normally difficult due to their geometric nature (Hannachi et al., 2007),
the high correlation between the first basis mode U1 with the mean TEC shows that U1 is
replicating the diurnal characteristics of the ionospheric TEC over Malindi. The fluctuations
observed in the higher order basis functions could be signatures of the different processes
(such as traveling thermospheric disturbances (TADs)) that influence the low latitude plasma
dynamics. Figure 1 (b) shows that the semiannual and annual variations in TEC have peaks
during the equinoxes and high solar activity years respectively. These coefficients are well
correlated with the solar radio flux measured at 10.7 cm wavelength (F10.7), confirming that
the main driver of ionospheric variability over Malindi is the changes in the extreme ultraviolet
(EUV) radiation from which the ionosphere owes its existence.
*3.2. Modeling of the coefficients*
Due to the rapid convergence of the basis functions, we used only the first six EOF modes
which accounted for 99.2% of the explained variance in the data to model the observed regional
TEC as derived from GNSS measurements at Malindi. The coefficients were modeled in terms
of harmonic functions with periods of 0, 0.5 and 1 year to represent the linear, semiannual and
annual variations respectively using equations 5−7.

$$C_j(d) = B_{j1}(d) + B_{j2}(d) + B_{j3}(d) \qquad (4)$$

where

$$B_{j1}(d) = a_{j1} + b_{j1}F10.7(d) + c_{j1}Dst(d) \qquad (5)$$


$$B_{j2}(d) = \left[a_{j2} + b_{j2}F10.7_{av}(d) + c_{j2}Dst(d)\right]\cos\left(\frac{2\pi d}{365.25}\right) + \left[d_{j2} + e_{j2}F10.7_{av}(d) + f_{j2}Dst(d)\right]\sin\left(\frac{2\pi d}{365.25}\right) \quad (6)$$


$$B_{j3}(d) = \left[a_{j3} + b_{j3}F10.7_{av}(d) + c_{j3}Dst(d)\right]\cos\left(\frac{4\pi d}{365.25}\right) + \left[d_{j3} + e_{j3}F10.7_{av}(d) + f_{j3}Dst(d)\right]\sin\left(\frac{4\pi d}{365.25}\right) \quad (7)$$





The daily averaged disturbance storm-time (Dst) and F10.7 indices used in the modeling process
were downloaded from Omniweb[1]. The $F10.7_{av}$ used in equations $5-7$ was calculated from
$F10.7_{av} = \frac{1}{2}(F10.7 + F10.7_{81})$, where $F10.7_{81}$ was the 81 day running average of F10.7. After
estimating the coefficients using a least squares fit to the GPS-derived TEC values measured
at Malindi, the modeled TEC was obtained using equation 1 by replacing the coefficients with
their modeled values. The variation of the observed GPS-derived TEC, the reconstructed TEC
from the first six EOF modes, and the modeled TEC are shown in Figure 2. It can be seen from
Figure 2 a & b that the six EOF modes were sufficient to reproduce the variation in the TEC.
Figure 2 c shows that the model quite well captured the diurnal, seasonal and the solar activity
variations in the observed TEC over Malindi. Correlation analysis between the observed and
the modeled TEC show a high positive correlation (Figure 2 d) with a correlation coefficient of
0.9225. This high correlation is an indication of the EOF decomposition method being capable
of reproducing the inherent features of the dynamic ionosphere at the crest of the anomaly
region.

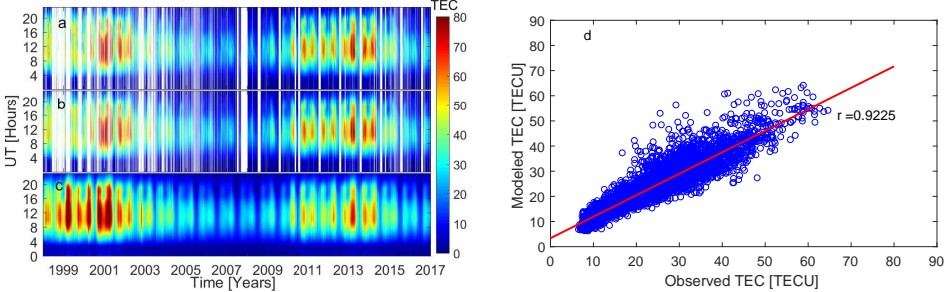

Figure 2: (a) GPS TEC, (b) reconstructed TEC and (c) modeled TEC over Malindi. (d) Correlation between EOF modeled TEC and GPS-derived TEC over Malindi from 1999 to 2017

### 3.3. Model validation

To asses the performance of the EOF-based TEC model, we compared the model results with
the TEC derived from IRI-2016 model and GIMs obtained from the website[2] of Center for Orbit
Determination in Europe (CODE). From here onwards, TEC from the EOF-based TEC model
will be referred to as EOF TEC, TEC derived from the GPS receiver in Malindi as GPS TEC,
TEC from CODE's GIMs as CODE's TEC and TEC from IRI-2016 model as IRI TEC. We
used both Kp and Dst to characterize the days into quiet and disturbed. A day was considered
to be quiet if Kp $\leq 3$.

### 3.3.1. Quiet days

Quiet days were selected from the equinox (March and September) and solstice (June and
December) months of high (2002), low (2009) and ascending (2011) solar activity phases in order
to validate the EOF-based TEC model. The data for the selected quiet days were excluded
from the matrix used to generate the model coefficients, and the same procedure for model
construction was repeated. The IRI TEC for the selected days were obtained from the web
interface of IRI-2016 model hosted at Omniweb[3]. To retrieve the IRI TEC, the location was
specified to coincide with the geographic coordinates of MAL2 GPS receiver and the topside

---

[1] https://omniweb.gsfc.nasa.gov/
[2] ftp://ftp.aiub.unibe.ch/CODE/
[3] https://omniweb.gsfc.nasa.gov/vitmo/iri2016_vitmo.html




boundary was set to its maximum value of 2000 km. The NeQuick option was used as the topside
electron density model and ABT-2009 for the bottom side thickness. Figure 3 shows hourly
diurnal variation of GPS TEC, IRI TEC, EOF TEC and CODE's TEC over Malindi for some
  selected quiet days. As expected, TEC values were higher in 2002, followed by 2011 and then

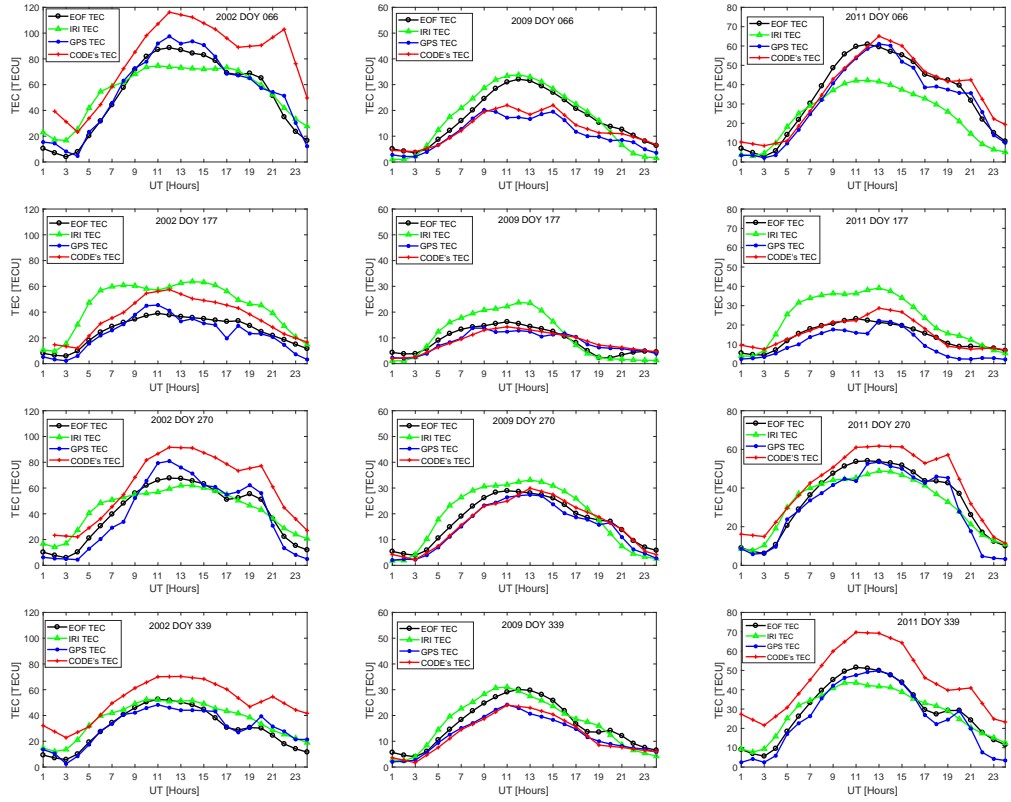

Figure 3: GPS TEC, IRI TEC, CODE's TEC and the EOF TEC over Malindi for some selected quiet days

2009. The IRI-2016 model overestimated the diurnal TEC over Malindi during the low solar
activity year 2009 and during winter solstice of all the years. The overestimate of the GPS TEC
by the IRI TEC is similar to what Olwendo et al. (2012) observed when they compared GPS
TEC measurements over Kenya with TEC from IRI-2007 model. During the equinox months of
higher solar activity years, IRI TEC values were higher and lower than the GPS TEC at about
3:00-9:00 UT and 11:00-13:00 UT respectively. The CODE's TEC overestimated the GPS TEC
especially during the higher solar activity years 2002 and 2011. The overestimate of the GPS
TEC by CODE's TEC was not much reflected during the low solar activity year 2009. The
EOF TEC on the other hand replicated the diurnal TEC quite well except on day of the year
(DOY) 066 and DOY 339 in 2009. In general, the highest correlation was observed between the
GPS TEC and CODE's TEC followed by the correlation of the GPS TEC with the EOF TEC.
It is worth noting that CODE's TEC mainly overestimated the TEC over Malindi especially
during higher solar activity years. Meanwhile, the IRI-2016 overestimated the GPS TEC over
MAL2 between 03:00 and 07:00 UT during high solar activity periods, and throughout the day
during lower solar activity years. This may be due to inadequate ingestion of ground based
data from the East African region in to the IRI model. As earlier mentioned, measurements





from ionosondes were used to provide ground data during IRI model construction, and such
data is currently limited over East Africa.
*3.3.2. Disturbed days*
To study the storm time performance of the EOF-based TEC model, we simulated TEC for
some selected geomagnetic storms. As stated earlier, the test days were excluded in the process
of generating the model coefficients. The IRI TEC for the storm days were obtained with
the storm model turned on. Figure 4 shows variation of the hourly diurnal TEC during some
   selected major geomagnetic storms. In Figure 4 a, TEC variation for a storm that occurred

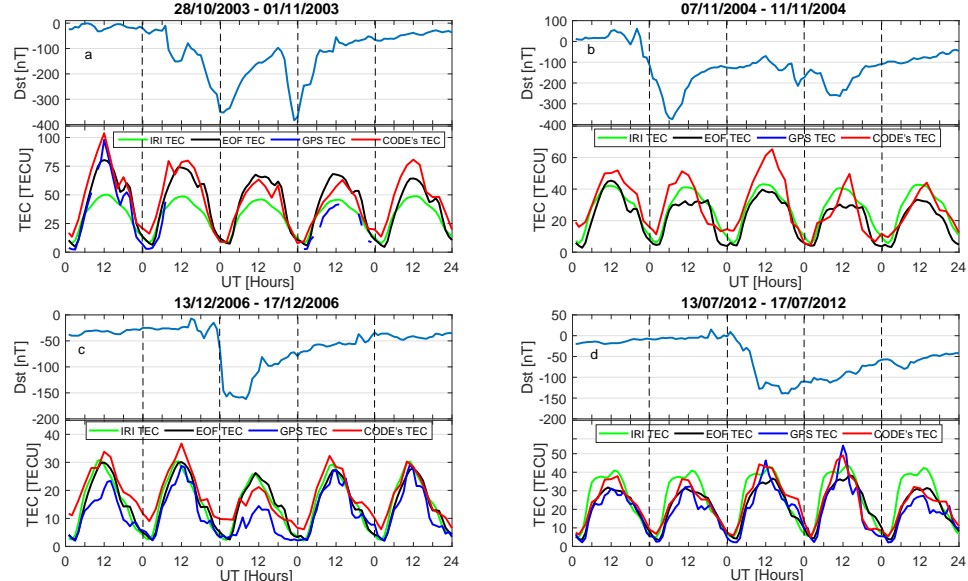

Figure 4: GPS TEC, IRI TEC, CODE's TEC and EOF TEC over Malindi for some selected storms. In these plots, the GPS TEC values are taken as a true representation of the ionosphere over Malindi. However, in the absence of GPS TEC, CODE's TEC were used to describe the ionospheric response to the storms

from 29-30 October 2003 are shown. No continuous GPS TEC measurements were available
from MAL2 IGS receiver during this storm period. As can be seen in CODE's TEC, the
storm had negative effect on the peak value of the TEC. The same negative storm effect was
replicated by the EOF and IRI models especially on 30-31 October 2003. Another major storm
occurred from 08-10 November 2004 with the main phase on the 08 November 2004 (Figure 4
b). Similarly, during this period, GPS TEC measurements were not available from MAL2 IGS
receiver. Both CODE's TEC and EOF TEC revealed negative storm effects on 08-November
and 10-November 2004 (Figure 4 b). However, the IRI TEC was not sensitive to the effect of the
geomagnetic storm. In Figure 4 c, the EOF TEC, CODE's TEC and IRI TEC showed negative
storm effects, consistent with the GPS TEC during the storm that occurred on 15-December
2006. A case of positive storm time effect on the ionosphere is shown in Figure 4 d where the
EOF TEC and CODE's TEC showed similar positive storm time effect as in the GPS TEC.
This was not reflected in the TEC derived from the IRI-2016 model.
*3.3.3. Statistical analysis*
From the EOF-based TEC model, we have simulated TEC for the months of March, June,
September and December for low (2009) and high (2013) solar activity years. In each of these





simulations, the data for the selected months were excluded from the data used to generate
the model coefficients. It is worthy noting that in generating the model coefficients, say for
March 2009, it is only the data of March 2009 that was excluded. The monthly median values
for the EOF TEC, IRI TEC and CODE's TEC were used to compute the root mean square
error (rmse) of the predicted TEC from the GPS TEC for every hour of the day. The diurnal
variation of the rmse values in 2009 and 2013 are shown in Figure 5. From Figure 5, the

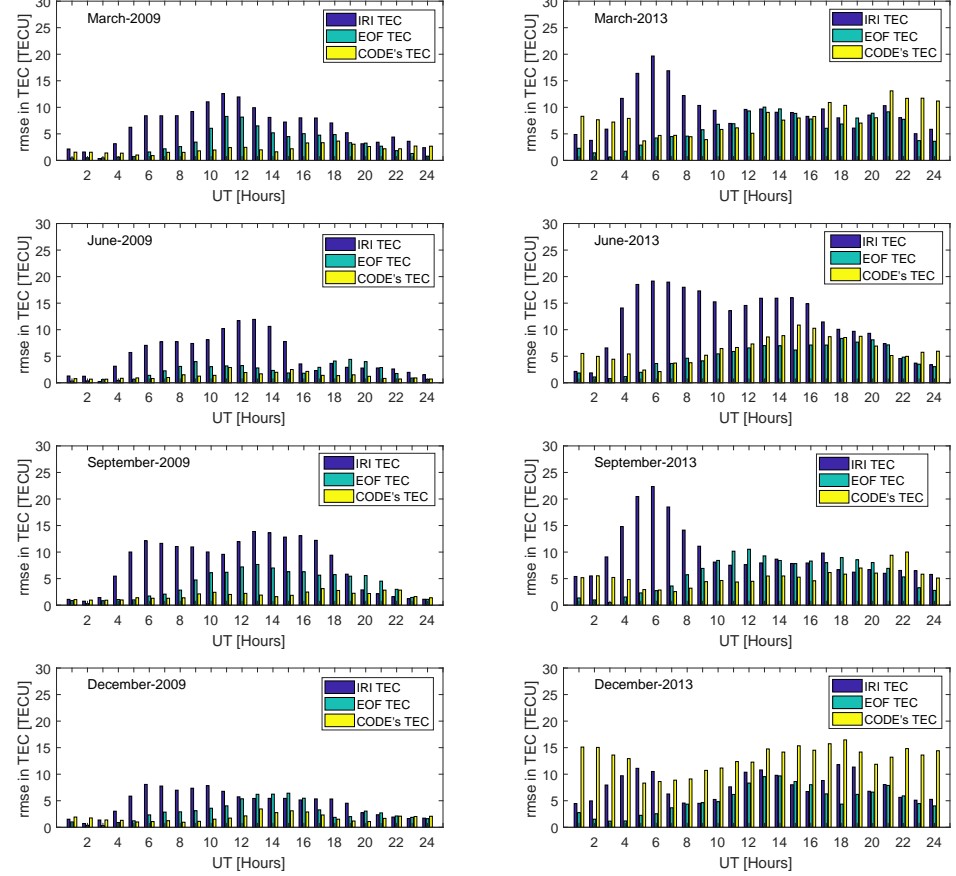

Figure 5: Diurnal variation of root mean square error (rmse) of IRI TEC, CODE's TEC and EOF TEC relative to GPS TEC over Malindi

uncertainty in predicting the observed TEC was higher in 2013 than in 2009. This could be
due to intensification of equatorial electrodynamic processes and the associated effects such as
TEC perturbations during higher solar activity years (Andima et al., 2015). These secondary
effects of the equatorial dynamo processes are not probably well captured by the models. A
comparison of the performance of the different models has shown that, CODE's GIMs predicted
the TEC over MAL2 with the least rmse values followed by the EOF model in 2009. The
greatest uncertainty in predicting the TEC over MAL2 in the same year was observed in the
IRI-2016 TEC. Similarly in 2013, the IRI TEC had higher rmse values compared to the EOF
TEC. CODE's GIMs showed the largest uncertainty in predicting the TEC over MAL2 during
December solstice of the high solar activity year 2013. The diurnal uncertainties in the ability
of IRI-2016 to predict the TEC over MAL2 exhibited two peaks in March, June and September



in 2009. The maximum rmse values in the IRI-2016 predicted TEC were observed between
11:00 and 13:00 UT in March and June, and 13:00-14:00 UT in September in 2009. December
solstice showed a single peak in the rmse values from 06:00-7:00 UT in 2009. Both the EOF
TEC and CODE's TEC had the largest rmse values from 11:00-14:00 UT in the same year. The
rmse values from 16:00-18:00 UT in March and September 2009 were quite higher than those
in June and December of the same year. In 2013, the rmse in the IRI-2016 predicted TEC in
the months of March and September had only single peaks which occurred at about 06:00 UT.
However, two peaks, one between 04:00 and 07:00 UT and second between 13:00 and 14:00 UT
were observed in June and December 2013. The smallest error in the IRI TEC and the EOF
TEC were observed after local midnight to about 03:00 UT in both 2009 and 2013.

## 4. Modeling TEC over African low latitudes

The first step in the regional TEC modeling was to extract the TEC for the African low
latitude region from CODE's GIMs. The daily GIMs were organized into bins of $2.5^O \times 5^O \times 1$ Hr
(latitude×longitude×LT). The binned data was then decomposed into the spatial and temporal
components according to the equation

$$\text{TEC(lat, lon, LT, m)} = \sum_{i=1}^{r} U_i(\text{lon, lat}) \times P_i(\text{LT, m}) \tag{8}$$

In equation 8, $U_i(\text{lon, lat})$ are the basis modes representing the spatial TEC variability and
$P_i(\text{LT, m})$ are the coefficients that describe the temporal TEC variations in terms of local time
(LT) and month (m). Figure 6 (a) shows the first four basis modes from the decomposition
in equation 8, and their corresponding expansion coefficients are shown in Figure 6 (b). The

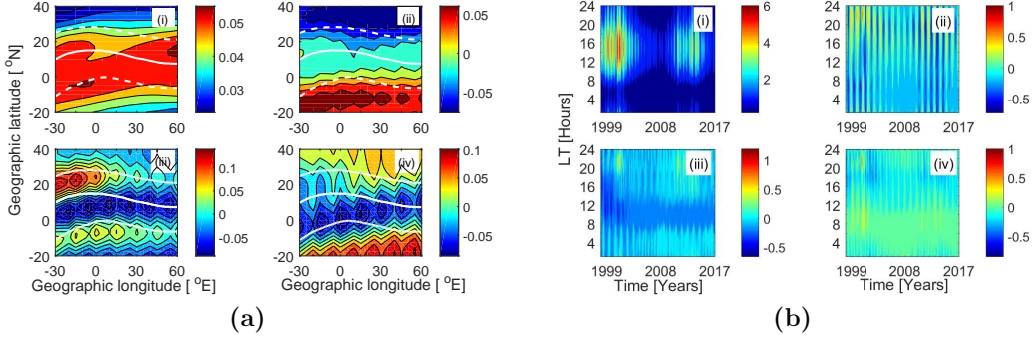

Figure 6: (a) The first four basis vectors from the first layer decomposition of the TEC data. (b) The coefficients
of the basis modes in (a)

first basis mode (Figure 6 (a) (i)) replicated the average TEC over African low latitude region
for the period 1999-2017, and therefore represents the average TEC characteristics over Africa.
This mode accounts for 97.012% of the TEC variability over the region. The coefficients of the
first basis mode show that TEC over African low latitudes exhibits, diurnal, seasonal and solar
activity dependences (Figure 6 (b) (i)). Shown in Figure 6 (a) (ii) is the second basis mode and
the coefficients of this mode are shown in Figure 6 (b) (ii). The second basis mode accounts
for 1.255% of the total TEC variability and it demonstrates the TEC variability in the two
hemispheres, possibly due to differences in the seasons in the two hemispheres. In Figure 6
(a) (iii), the effect of the magnetic and electric field coupling resulting in the EIA appears to
be evident. This mode accounts for 0.455% of the total TEC variability over the African low
latitude region.





The temporal component in equation 8 was further broken into the diurnal and long-term
(seasonal, annual and solar cycle) variations by another decomposition which we refer to here
as the second layer decomposition expressed as

$$P_i(LT, d) = \sum_{i,j=1}^{m} U_{i,j}(LT) \times A_{i,j}(d). \tag{9}$$

In equation 9, $U_{i,j}(LT)$ are the basis functions of the $i^{th}$ first level coefficients and $A_{i,j}(d)$
are the coefficients of $U_{i,j}(LT)$. Figure 7 (a) show the first basis modes for each of the first
four expansion coefficients in the first level decomposition. Equations 5 - 7 were then used to

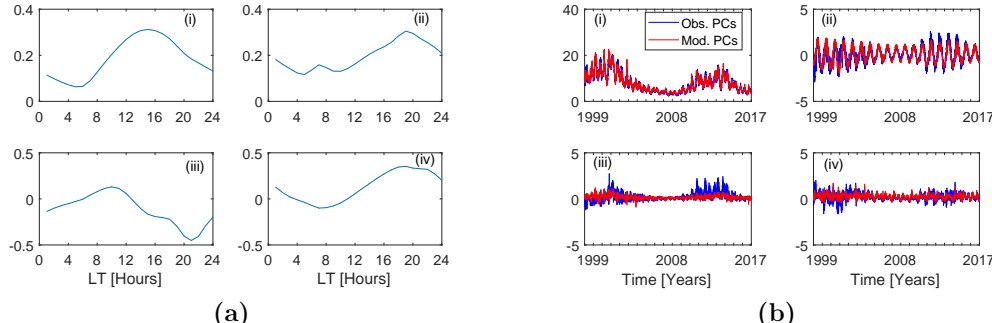

Figure 7: (a) The first basis modes of the first four first layer expansion coefficients. (b) The expansion coefficients (blue) and their modeled values (red) for the basis modes in (a).

model the coefficients $A_{i,j}(d)$. Figure 7 (b) shows the coefficients $A_{i,j}(d)$ together with their
model predicted values. Using the modeled values of the coefficients, the regional TEC was
then reconstructed in a reverse order. We first used equation 9 to obtain the coefficients for
the first layer decomposition and then applied equation 8 to determine the TEC in each grid
cell. Figure 8 shows the modeled TEC, CODE's TEC and IRI TEC for DOY 070 in 2015. It
can be seen from Figure 8 that the model has quite well reproduced the main features of the
EIA region. Higher correlations are observed between EOF modeled TEC and CODE's TEC
(Figures 8 a & b). This high correlation is an indication that the model predicted results could
offer a good alternative in estimating background TEC since CODE's TEC are derived from
GNSS measurements.

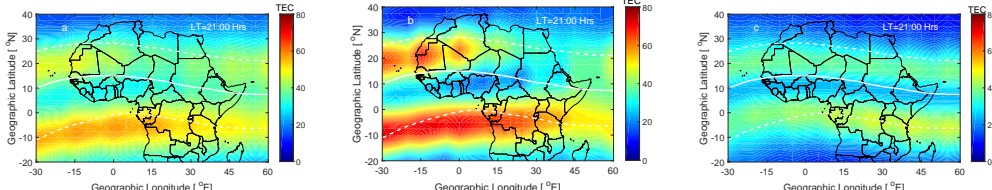

Figure 8: EOF TEC (a), CODE's TEC (b) and the IRI TEC (c) for the DOY 070 in 2015. Shown on the plots are the local times in hours


## 5. TEC trends

In the past, few studies (e.g Lean et al. 2011; Lastovicka et al. 2017) have derived long-term TEC
trends, and these mainly used TEC data from GIMs. In this work, we have used GPS-derived





TEC and GIMs to study TEC trends over the low latitude region of Africa. A key aspect
in trend studies is the art of suppressing the solar and magnetic activity influences on these
trends. We used the EOF modeled TEC as background TEC to remove the solar and magnetic
effects in influencing the TEC trends. First, the monthly median TEC values were calculated
from the daily TEC. The median TEC values were then modeled using similar equations as in
equations 5−7. In these equations, the daily inputs were replaced with their monthly averages.
The modeled TEC values were then subtracted from the monthly medians to obtain the TEC
residuals ($\Delta$TEC). The trend was then determined using the equation (Lastovicka et al., 2006;
Bremer et al., 2012).

$$\Delta\text{TEC} = A + B*\text{time (year)} \tag{10}$$

where A is the constant part and B the slope (trend) of the time-dependent TEC residuals.
Lastovicka et al. (2017) attributed the positive global TEC trends reported in Lean et al.
(2011) to lower TEC values in CODE's GIMs especially prior to 2003. To test this assertion,
we have estimated the long-term trends in TEC for the periods 1999-2017 and 2003-2017
using GPS-derived TEC, CODE's TEC and TEC from IGS GIMs. Figure 9 shows the TEC

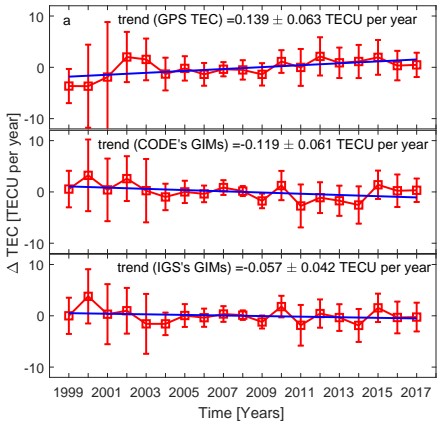
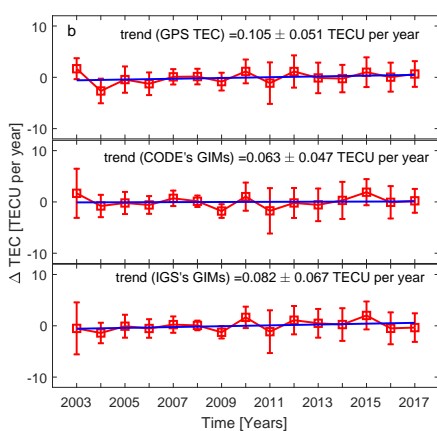

Figure 9: The yearly median residual TEC after removing the solar and magnetic activity variations for the periods 1999-2017 (a) and 2003-2017 (b). The upper panels are for the TEC derived from MAL2 IGS receiver, the middle panels are for TEC from CODE's GIMs and the lower panels are for TEC from IGS's GIMs corresponding to the location of MAL2 IGS receiver. The straight lines in these plots are the first degree polynomial fits used to estimate the trends

trends obtained from GPS-derived TEC and TEC from GIMs of CODE and IGS over Malindi.
Trends of 0.139±0.063 TECU/year, -0.119±0.061 TECU/year and -0.057±0.042 TECU/year
were obtained using the GPS TEC, CODE's GIMs and IGS GIMs over Malindi respectively.
Though the trend values in Figure 9a were slightly different, their 95% confidence bounds reveal
a slight positive TEC trend for the period 1999-2017. For the period 2003-2017, the trend
estimates from the three data sets show that TEC trends over MAL2 are positive. To study
the trends in TEC over the African low latitude region, we used the GIMs and estimated the
trends in each of the $2.5^O \times 5^O$ (latitude×longitude) grids. The trends are shown in Figure 10(a)
and (c) for the period 1999-20017 and in Figure 10(b) and (d) for the period 2003-2017. The
trends in Figure 10 show a latitudinal dependence with the trends in the vicinity of the crest
of the EIA region being more positive than those near the magnetic equator, where the trend
was negative over most of the African equatorial region. Analysis of the data from 1999-2017





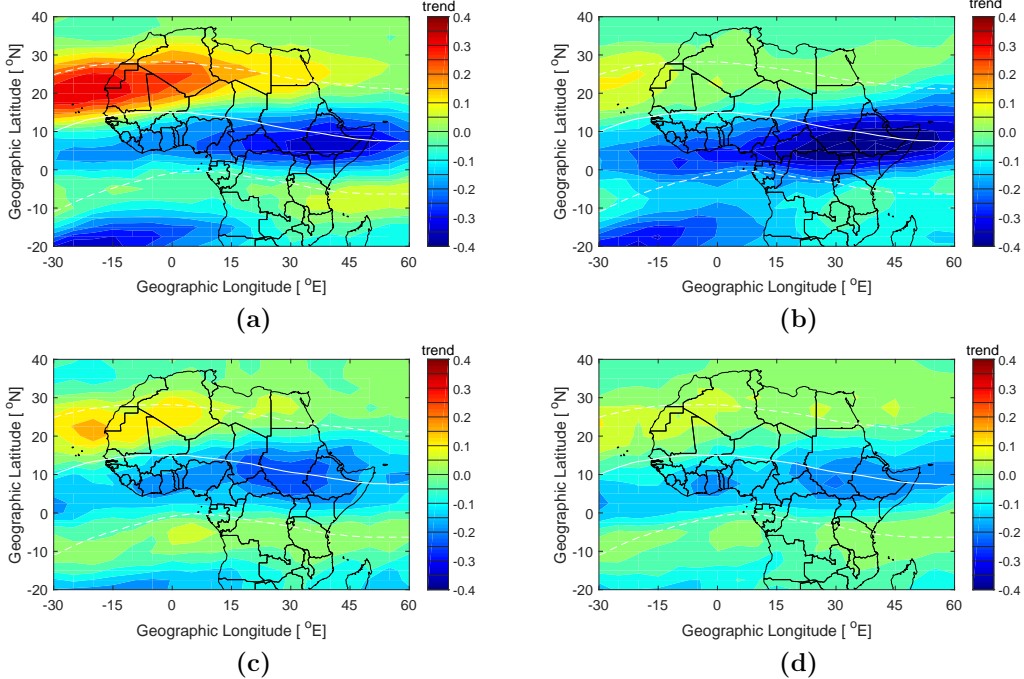

Figure 10: Trends in TEC over the African low latitude region derived from CODE's ((a) and (b)) and IGS's ((c) and (d)) GIMs. (a) and (c) are for the period 1999-2017 while (b) and (d) are for the period 2003-2017

revealed higher average value of TEC trends than that from 2003-2017 for both CODE's and IGS GIMs, though the general pattern of the trends remained unchanged. The trend pattern observed in this study confirm the latitudinal variation in TEC trends reported in Lean et al. (2011). The difference in the trend magnitudes for the periods 1999-2017 and 2003-2017 could be as pointed out earlier due to a bias towards lower values of GIMs prior to 2002 (Lastovicka et al., 2017; Emmert et al., 2017). If so, the trends in TEC over African low latitudes are therefore mainly negative, though cases of slight positive trends are also possible especially at the crest of the EIA region. Geomagnetic or anthropogenic factors are often the plausible physical mechanisms to explain trends in upper atmospheric parameters. The anthropogenic contributions to trends arises through the accumulation of greenhouse gases which result in decrease of atomic oxygen in the upper atmosphere. Some authors (e.g Danilov and Mikhailov 1999; Mikhailov 2008) attribute the geomagnetic control of the trend to geomagnetic storms that result in compositions changes which in turn affect ionospheric currents. This latter view may explain the latitudinal dependence in the trends over African low latitude region.

## 6. Summary and Conclusion

We have used EOF expansion together with least square regression to model TEC over the African low latitude region. We first developed a single station model over MAL2, a station at the southern crest of the EIA, and then constructed a regional model to predict TEC over the African low latitudes. Despite the complicated nature of the low latitude ionosphere, the model over MAL2 was able to satisfactorily reproduce the diurnal, seasonal and solar activity variations in the observed TEC. Comparison of the model results with IRI-2016 derived TEC showed that the EOF-based TEC model was more accurate in predicting the daytime TEC over Malindi than IRI-2016. On the other hand, CODE's GIMs were better correlated with GPS





TEC than the TEC from the EOF-based model, though CODE's GIMs mainly overestimated the GPS TEC over MAL2 during higher solar activity years. The large discrepancy of the IRI-2016 predicted daytime TEC from the observed TEC over MAL2 during periods of low solar activity and during the winter solstice could be due to over representation of the effects of the low latitude E×B plasma drifts and thermospheric winds in the model. The regional model reproduced quite well the known features of the low latitude ionosphere. Using the TEC from the EOF-based model as background TEC to suppress solar and magnetic activity dependence of TEC, we estimated trends in TEC over the African low latitude region. The regional trends showed a latitudinal dependence with the trends in the vicinity of the magnetic equator being more negative than those at the crest of the EIA.

**Acknowledgement**

This study was made possible by financial support from International Science Programme (ISP) of Uppsala University in Sweden. We acknowledge the administration and staff of the Space Science Directorate of the South African National Space Agency (SANSA) for the support during the research visit of the first author to the institution.

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
