# Peer review of "Ann. Geophys. Discuss., https://doi.org/10.5194/angeo-2018-87"

_Annales Geophysicae, 2018_

## Referee Comment (RC1) · Anonymous Referee #1 · 8 Aug 2018

**Comments:**

In this manuscript, the authors present an investigation of the total electron content (TEC) at a single station and low latitude region over Africa. They applied empirical orthogonal function (EOF) to model the seasonal and solar cycle and storm-time variations of TEC and to further determine the trend of TEC over the data period.

As the features of the ionosphere over Africa is relatively less studied, the work is possible for publication in AG after reasonable modifications as listed below.

**Detailed comments/suggestions:**

1. Equation (1): Did it require to include a mean term?

2. Page 3: Line 37: I cannot see the secondary maximum at about 20:00 UT from The figure.

3. Figure 1: What is the red curve? No words are given.

4. Page 4, Line 16: The statement is incorrect. Periods of 0 is not the linear variation.

5. Equation (4): Why such presentation is reasonable or enough. No words to support it, no references are cited.

6. Equation (5): Same to Equation (4). Moreover, (1) F10.7 in Equation (5), but F10.7av in Equation (6). (2) Is it enough to present the geomagnetic activity condition with Dst? And it is a linear relationship? Without any time delay? (3) How to simulate/reproduce the solar effect of TEC is essential for successful

model TEC and further estimate the trend of TEC. In this work, the solar index uses F10.7 av. It did not provide references (suggest to cite two key works, *Richards et al*., 1994: Richards, P. G., J. A. Fennelly, and D. G. Torr (1994), EUVAC: A solar EUV flux model for aeronomic calculations, *J. Geophys Res*., 99, 8981-8992; Liu et al., 2006: Liu, L., W. Wan, B. Ning, et al. (2006), Solar activity variations of the ionospheric peak electron density, *J. Geophys. Res*., 111, A08304, doi:10.1029/2006JA011598). (4) According to the investigation of Liu and Chen (2009) and Liu et al. (2009), the TEC is better to present as a second-order polynomial with F10.7, especially under the situation for estimating trend. References: Liu, L., and Y. Chen (2009), Statistical analysis on the solar activity variations of the TEC derived at JPL from global GPS observations, *J. Geophys. Res*., 114, A10311, doi:10.1029/2009JA014533.

Liu, L., W. Wan, B. Ning, and M.-L. Zhang (2009), Climatology of the mean TEC derived from GPS Global Ionospheric Maps, *J. Geophys. Res*., 114, A06308, doi:10.1029/2009JA014244

7. Figure 2: An equation is welcome to give the regression. I am curious as why there is a significant intercept.

8. Figure 4: The case on 07/11/2004 has no observation of GPS-TEC. If so, what is the value to include it here.

9. Figure 6: What are the white curves? So are those in Figures 8 and 10.

10. Page 12: Line 5: Since the processing of TEC may introduce biases to TEC, my question is how about the possible influence of your TEC on the trend?

11. Figure 10: An issue is the influence of different presentation of solar dependency of TEC, linear or higher order on the trend. According to the investigation of Liu and Chen (2009) and Liu et al. (2009), the TEC shows saturation in equatorial regions, such a linear presentation as in this work will introduce what influence on the estimated trend?

12. Page 11, Lines 18-21: The trend is positive. In contrast, in other works it is negative. What causes the difference?

13. According to Equation (1) and Equation (4), there is expected that we can estimate the trend as different local times. My next question is how about the local time variation of the trend of TEC?

---

## Author Comment (AC1) · 3 Sep 2018

**Response to Reviewer#RC1**

**Comment 1**

Equation (1): Did it require to include a mean term?

**Response**

Thank you for the inquiry. Singular Value Decomposition factorizes a matrix  $M_{m \times n}$  in to three other matrices U, S and V as given in Equation 2. To recover  $M_{m \times n}$ , we take the matrix product of U, S and V. The  $\sum_{j=1}^{n}$  in Equation 1 is for the sum of the product of the matrix entries in the multiplication of U by C, where C is given by Equation 2. Therefore there is no mean term in Equation 1

**Comment 2**

Page 3: Line 37: I cannot see the secondary maximum at about 20:00 UT from the Figure

**Response**

Thank you for the observation. Critical analysis of Figure 1 (a) shows that there is no secondary maxima at about 20:00 UT as we had earlier stated. What appears at about 20:00 UT is an enhancement. The sentence was therefore rephrased to: Figure 1 (a) shows that the average diurnal TEC (red curve) over Malindi has a pre-dawn minimum at about 3:00 UT, a maximum at about 11:30 UT and an enhancement from 18:00 to 20:00 UT. The maximum at 11:30 UT is possibly due to increased ionization as the solar zenith angle is nearly zero over Malindi around this time. The post sunset increase in the TEC from 18:00 to 20:00 UT could be due to an enhancement in the eastward electric field before its westward reversal at night.

**Comment 3**

Figure 1: What is the red curve? No words are given.

**Response**

Thank you for the observation. What the red curve represents has now been clearly stated in the caption. The caption of Figure 1 was therefore changed to:

The first six basis functions (a) representing the diurnal variation and their coefficients (b) which show the long-term variation of TEC over MAL2. The red curve in (a) and (b) show the diurnal variation of average TEC from 1999-2017 over MAL2 and the daily solar radio flux measured at 10.7 cm wavelength (F10.7) respectively.

**Comment 4**

Page 4, Line 16: The statement is incorrect. Periods of 0 is not the linear variation.

**Response**

We acknowledge that a linear function is not of period zero. In fact, a linear function has infinite period. We have deleted the incorrect statement and then updated the sentence as follows.

For an effective TEC model, the choice of parameters to model the solar and magnetic activity influences on the TEC is important. Based on our observations in Table RC1, it was reasonable to use F10.7av and Dst indices to model the solar and magnetic dependences of TEC over Malindi. We then expressed the EOF coefficients as a sum of linear and harmonic functions following the procedure of Zhang *et al.* (2009) as

$$\mathrm{C}_{j}(\mathrm{d}) = \mathrm{B}_{j1}(\mathrm{d}) + \mathrm{B}_{j2}(\mathrm{d}) + \mathrm{B}_{j3}(\mathrm{d})$$

The linear term  $B_{j1}(d)$  is to account for the linear variation of the EOF coefficients with solar and magnetic activities and is given by

$$\mathrm{B}_{j1}(\mathrm{d}) = \mathrm{a}_{j1} + \mathrm{b}_{j1}\mathrm{F10.7}_{av}(\mathrm{d}) + \mathrm{c}_{j1}\mathrm{Dst}(\mathrm{d})$$

The semiannual and annual variations in the EOF coefficients are represented in equation 1 by the harmonic terms  $B_{i2}(d)$  and  $B_{i3}(d)$  of periods 0.5 and 1 year (365.25 days) respectively as

$$B_{j2}(d) = \left[a_{j2} + b_{j2}F10.7_{av}(d) + c_{j2}Dst(d)\right] \cos\left(\frac{2\pi d}{365.25}\right) + \left[d_{j2} + e_{j2}F10.7_{av}(d) + f_{j2}Dst(d)\right] \sin\left(\frac{2\pi d}{365.25}\right)$$
(1)

$$B_{j3}(d) = \left[a_{j3} + b_{j3}F10.7_{av}(d) + c_{j3}Dst(d)\right] \cos\left(\frac{4\pi d}{365.25}\right) + \left[d_{j3} + e_{j3}F10.7_{av}(d) + f_{j3}Dst(d)\right] \sin\left(\frac{4\pi d}{365.25}\right)$$
(2)

**Comment 5**

Equation (4): Why such presentation is reasonable or enough. No words to support it, no references are cited.

**Response**

We have provided a reference in support of the method.

**Comment 6 a**

Equation (5): Same to Equation (4). Moreover, (1) F10.7 in Equation (5) and F10.7av in Equation (6)

**Response**

Thank you for the observation. In equation (5) it should have been F10.7av not F10.7. This was a typing error and has been corrected to:

$$B_{j1}(d) = a_{j1} + b_{j1}F10.7av(d) + c_{j1}Dst(d)$$

**Comment 6 b**

Is it enough to present the geomagnetic activity condition with Dst?

**Response**

(i) The Dst is based on measurements at near equatorial magnetic observatories (Honolulu, San Juan, Hermanus and Kakioka) and therefore it predominantly gives an indication of the equatorial ring current variations. Therefore Dst is more appropriate than AE since the observations are equatorial rather than auroral.

|    | Solar indices       |                     |                     | Magnetic indices |            |              |
|----|---------------------|---------------------|---------------------|------------------|------------|--------------|
|    | SSN                 | F10.7               | $F10.7_{av}$        | Кр               | AE         | Dst          |
| C1 | 71.2( 71.6 ) | 75.8( 76.0 ) | 79.1( 79.2 ) | 19.0(18.8)       | 15.6(15.1) | -23.1(-23.0) |
| C2 | 36.6( 37.0 ) | 35.4(35.2)          | 37.2(37.1)          | 9.5(8.7)         | 12.1(10.0) | -5.5(-6.0)   |
| C3 | 16.9(17.4)          | 16.7(17.2)          | 16.4(16.7)          | 4.2(7.2)         | 4.6(9.1)   | -4.1(-1.6)   |
| C4 | 4.4(4.8)            | 7.5(8.0)            | 8.1(8.5)            | 17.8(14.4)       | 20.1(18.4) | -13.3(-4.0)  |
| C5 | 5.8(5.9)            | 6.9(6.9)            | 7.0(7.0)            | 2.3(0.3)         | 1.3(0.2)   | -6.8(-6.5)   |
| C6 | 0.2(0.2)            | 1.9(1.6)            | 1.8(1.6)            | 1.8(0.7)         | 3.9(1.8)   | -4.4(-3.1)   |

Table RC1: Correlation of the expansion coefficients with the different solar and magnetic indices. In brackets are the coefficients at a lag of one day

(ii) The Dst and Kp indices are not independent, so the best of these would be the one that gives the highest correlation with the TEC observations. Of the three magnetic indices considered, it is Dst that showed the highest correlation with the coefficients of the first EOF mode which explained 96.8% of the TEC variance over MAL2.

**Comment 6 c**

Is the linear relation between TEC and Dst sufficient?

**Response**

Numerous studies have used linear terms of solar and magnetic indices (eg Lastovicka et al.(2017), Uwamahoro and Habarulema (2015), Ercha et al 2012, Zhang et al 2009) to model TEC. More so, our data set did not reveal either a second or higher order relation between the daily averaged TEC and Dst, but rather a weak linear relation. Based on the observations with our data set, we adopted the linear relation in our model

**Comment 6 d**

Is there no time delay between the occurrence of a geomagnetic storm and its effect on the ionosphere?

**Response**

Scherlies and Fejer (1997) observed that the equatorial plasma responds to high latitude current disturbance in time scales of 1-12 hrs (short-term effect) and 20-30 hrs (long-term effect). Owing to the fact that we are using daily averages of Dst, the short term effect can not be considered in the modeling process. To investigate the long term effects, we determined the correlation between the EOF coefficients and the magnetic indices at zero and one day lags. Our correlation analysis (see Table RC1) did not reveal significant differences in the correlation coefficients at zero and one day lags. In fact the daily Dst values showed slightly higher correlation coefficients at zero lag with the EOF coefficients except for the second EOF mode. Based on these observations, we used the daily Dst values with zero lag to model the EOF coefficients. As mentioned earlier, such a representation has also been used by many authors (eg Latovicka et al 2017, Uwamahoro and Habarulema (2015), Ercha et al 2012, Zhang et al 2009) to model TEC

**Comment 6 e**

How to simulate/reproduce the solar effect of TEC is essential for successful model TEC and further estimate the trend of TEC. In this work, the solar index uses F10.7 av. It did not provide references (suggest to cite two key works, Richards et al., 1994: Richards, P. G., J. A. Fennelly, and D. G. Torr (1994), EUVAC: A solar EUV flux model for aeronomic calculations,

J. Geophys Res., 99, 8981-8992; Liu et al., 2006: Liu, L., W. Wan, B. Ning, et al. (2006), Solar activity variations of the ionospheric peak electron density, J. Geophys. Res., 111, A08304, doi:10.1029/2006JA011598).

**Response**

Thank you for the suggestions. Appropriate references were cited as suggested

**Comment 6 f**

According to the investigation of Liu and Chen (2009) and Liu et al. (2009), the TEC is better to present as a second-order polynomial with F10.7, especially under the situation for estimating trend. References: Liu, L., and Y. Chen (2009), Statistical analysis on the solar activity variations of the TEC derived at JPL from global GPS observations, J. Geophys. Res., 114, A10311, doi:10.1029/2009JA014533. Liu, L., W. Wan, B. Ning, and M.-L. Zhang (2009), Climatology of the mean TEC derived from GPS Global Ionospheric Maps, J. Geophys. Res., 114, A06308, doi:10.1029/2009JA014244

**Response**

Thank you for the suggestion. We investigated the relationship between the daily mean TEC

Figure RC1: Scatter plots for (a) daily mean TEC and F10.7 (b) daily mean TEC and F10.7av

and F10.7av. As suggested by Liu et al. (2009), representing F10.7 as second order polynomial has smaller root mean square error (see Figure RC1). The difference in rmse in using a linear fit and that of second order polynomial appears to be small especially with F10.7av.

We further proceeded to model the TEC by representing F10.7av as second order polynomial using the equations below.

$$C_{i}(d) = B_{i1}(d) + B_{i2}(d) + B_{i3}(d)$$

where

$$\mathrm{B}_{j1}(\mathrm{d}) = \mathrm{a}_{j1} + \mathrm{b}_{j1}\mathrm{F10.7}_{av}(\mathrm{d}) + \mathrm{c}_{j1}[\mathrm{F10.7}_{av}(\mathrm{d})]^2 + \mathrm{d}_{j1}\mathrm{Dst}(\mathrm{d})$$

 $B_{j2}(d) = \left[a_{j2} + b_{j2}F10.7_{av}(d) + c_{j2}[F10.7_{av}(d)]^2 + d_{j2}Dst(d)\right] \cos\left(\frac{2\pi d}{365.25}\right) \\ + \left[e_{j2} + f_{j2}F10.7_{av}(d) + g_{j2}[F10.7_{av}(d)]^2 + h_{j2}Dst(d)\right] \sin\left(\frac{2\pi d}{365.25}\right) \\ + \left[e_{j2} + f_{j2}F10.7_{av}(d) + g_{j2}[F10.7_{av}(d)]^2 + h_{j2}Dst(d)\right] \sin\left(\frac{2\pi d}{365.25}\right) \\ + \left[e_{j2} + f_{j2}F10.7_{av}(d) + g_{j2}[F10.7_{av}(d)]^2 + h_{j2}Dst(d)\right] \sin\left(\frac{2\pi d}{365.25}\right) \\ + \left[e_{j2} + f_{j2}F10.7_{av}(d) + g_{j2}[F10.7_{av}(d)]^2 + h_{j2}Dst(d)\right] \sin\left(\frac{2\pi d}{365.25}\right) \\ + \left[e_{j2} + f_{j2}F10.7_{av}(d) + g_{j2}[F10.7_{av}(d)]^2 + h_{j2}Dst(d)\right] \sin\left(\frac{2\pi d}{365.25}\right) \\ + \left[e_{j2} + f_{j2}F10.7_{av}(d) + g_{j2}[F10.7_{av}(d)]^2 + h_{j2}Dst(d)\right] \sin\left(\frac{2\pi d}{365.25}\right) \\ + \left[e_{j2} + f_{j2}F10.7_{av}(d) + g_{j2}[F10.7_{av}(d)]^2 + h_{j2}Dst(d)\right] \sin\left(\frac{2\pi d}{365.25}\right) \\ + \left[e_{j2} + f_{j2}F10.7_{av}(d) + g_{j2}[F10.7_{av}(d)]^2 + h_{j2}Dst(d)\right] \sin\left(\frac{2\pi d}{365.25}\right) \\ + \left[e_{j2} + f_{j2}F10.7_{av}(d) + g_{j2}[F10.7_{av}(d)]^2 + h_{j2}Dst(d)\right] \sin\left(\frac{2\pi d}{365.25}\right) \\ + \left[e_{j2} + f_{j2}F10.7_{av}(d) + g_{j2}[F10.7_{av}(d)]^2 + h_{j2}Dst(d)\right] \sin\left(\frac{2\pi d}{365.25}\right) \\ + \left[e_{j2} + f_{j2}F10.7_{av}(d) + g_{j2}F10.7_{av}(d)\right] + \left[e_{j2} + f_{j2}F10.7_{av}(d)\right] \\ + \left[e_{j2} + f_{j2}F10.7_{av}(d) + g_{j2}F10.7_{av}(d)\right] + \left[e_{j2} + f_{j2}F10.7_{av}(d)\right] \\ + \left[e_{j2} + f_{j$

 $B_{j3}(d) = \left[a_{j3} + b_{j3}F10.7_{av}(d) + c_{j3}[F10.7_{av}(d)]^2 + d_{j3}Dst(d)\right] \cos\left(\frac{4\pi d}{365.25}\right) + \left[e_{j3} + f_{j3}F10.7_{av}(d) + g_{j3}[F10.7_{av}(d)]^2 + h_{j3}Dst(d)\right] \sin\left(\frac{4\pi d}{365.25}\right) + \left[e_{j3} + f_{j3}F10.7_{av}(d) + g_{j3}[F10.7_{av}(d)]^2 + h_{j3}Dst(d)\right] \sin\left(\frac{4\pi d}{365.25}\right) + \left[e_{j3} + f_{j3}F10.7_{av}(d) + g_{j3}[F10.7_{av}(d)]^2 + h_{j3}Dst(d)\right] \sin\left(\frac{4\pi d}{365.25}\right) + \left[e_{j3} + f_{j3}F10.7_{av}(d) + g_{j3}[F10.7_{av}(d)]^2 + h_{j3}Dst(d)\right] \sin\left(\frac{4\pi d}{365.25}\right) + \left[e_{j3} + f_{j3}F10.7_{av}(d) + g_{j3}[F10.7_{av}(d)]^2 + h_{j3}Dst(d)\right] \sin\left(\frac{4\pi d}{365.25}\right) + \left[e_{j3} + f_{j3}F10.7_{av}(d) + g_{j3}[F10.7_{av}(d)]^2 + h_{j3}Dst(d)\right] \sin\left(\frac{4\pi d}{365.25}\right) + \left[e_{j3} + f_{j3}F10.7_{av}(d) + g_{j3}[F10.7_{av}(d)]^2 + h_{j3}Dst(d)\right] \sin\left(\frac{4\pi d}{365.25}\right) + \left[e_{j3} + f_{j3}F10.7_{av}(d) + g_{j3}[F10.7_{av}(d)]^2 + h_{j3}Dst(d)\right] \sin\left(\frac{4\pi d}{365.25}\right) + \left[e_{j3} + f_{j3}F10.7_{av}(d) + g_{j3}[F10.7_{av}(d)]^2 + h_{j3}Dst(d)\right] \sin\left(\frac{4\pi d}{365.25}\right) + \left[e_{j3} + f_{j3}F10.7_{av}(d) + g_{j3}F10.7_{av}(d)\right] + \left[e_{j3} + g_{j3}F10.7_{av}(d) + g_{$

Figure RC2: Scatter plots for observed TEC and modeled TEC using (a) F10.7av as a linear term (b) F10.7av as a second order polynomial

Figure RC2 shows that representing the solar activity as a second order polynomial results in smaller error than using the linear relationship. However, the difference in the rmse appears insignificant and we think will not result in any substantial differences in the conclusions

**Comment 7**

Figure 2: An equation is welcome to give the regression. I am curious as to why there is a significant intercept.

**Response**

A regression equation has been included in Figure 2 (d) as shown in Figure RC3. A positive

---

## Referee Comment (RC2) · Anonymous Referee #2 · 3 Oct 2018

**Review of the manuscript: Modeling of GPS total electron content over the African low latitude region using empirical orthogonal functions**

Authors: Geoffrey Andima , Emirant Amabayo, Edward Jurua, Pierre J Cilliers.

The paper presents empirical orthogonal function (EOF) based models of total electron content based on GPS TEC and GIM data, as well as a study of trends in TEC over low latitude region of the African sector. The contribution of the paper towards research progress is not clear and I am inviting the author to revise it taking into consideration the following points: 1) Specific objective of the paper, 2) Contribution of the paper to research progress, 3) specification of the new finding in this work. As it is difficulty for the reviewer to evaluate the contribution of this work to already existing works, I recommend a major revision.

**Major comments**

Comment 1: EOF analysis has been applied to TEC/$foF_2$ modelling in low and middle latitudes, during geomagnetically quiet and storm conditions, and at regional/global scale (References below).

[1]. A, E., D. Zhang, A. J. Ridley, Z. Xiao, and Y. Hao (2012), A global model: Empirical orthogonal function analysis of total electron content 1999 - 2009 data, J. Geophys. Res., 117, A03328, doi:10.1029/2011JA017238.

[2]. Dabbakuti, J. R. K., & Ratnam, D. V. (2017). Modeling and analysis of GPS-TEC low latitude climatology during the 24th solar cycle using empirical orthogonal functions. Advances in Space Research, 60(8), 1751 - 1764.

[3]. Dabbakuti, J. R. K., & Ratnam, D. V. (2016). Characterization of ionospheric variability in TEC using EOF and wavelets over low-latitude GNSS stations. Advances in Space Research, 57(12), 2427 - 2443.

[4].Mao, T., W. Wan, X. Yue, L. Sun, B. Zhao, and J. Guo (2008), An empirical orthogonal function model of total electron content over China, Radio Sci., 43, RS2009, doi:10.1029/2007RS003629.

[5]. Uwamahoro, J. C., and J. B. Habarulema (2015), Modelling total electron content during geomagnetic storm conditions using empirical orthogonal functions and neural networks, J. Geophys. Res. Space Physics, 120, 11,000 - 11,012, doi:10.1002/2015JA021961.

[6]. Ercha, A., D.-H. Zhang, Z. Xiao, Y.-Q. Hao, A. J. Ridley, and M. Moldwin (2011), Modeling ionospheric f o F 2 by using empirical orthogonal function analysis, Ann. Geophys., 29(8), 1501 - 1515.

[7]. Chen, Z., S.-R. Zhang, A. J. Coster, and G. Fang (2015), EOF analysis and modeling of GPS TEC climatology over North America, J. Geophys. Res. Space Physics, 120, 3118 - 3129, doi:10.1002/2014JA020837.

[8]. Le, H., N. Yang, L. Liu, Y. Chen, and H. Zhang (2017), The latitudinal structure of nighttime ionospheric TEC and its empirical orthogonal functions model over North American sector, J. Geophys. Res. Space Physics, 122, 963977, doi:10.1002/2016JA023361.

In my point of view, the presented work seems to be a repetition of what has been done previously and its contribution towards research progress referring to existing works is not clear. I am suggesting the authors to revise the introduction and include the references provided above and then highlight briefly and concisely their contribution to research progress.

Comment 2: (Page 2, lines: 18 - 19). The authors mention that 2-hour GIM data was interpolated to 1-hour data and it is evident that during interpolation some errors are introduced. I am suggesting the authors to clarify how the interpolation method used in this study has been validated before being applied, and how errors due to interpolation will affect TEC modelling results.

Comment 3: Discussion of the results and conclusions should be revised. The authors should highlight the main findings of the current work referring to previous works.

**Minor comments**

Comment 1: (Page 3, line 33) The six EOF modes are not elements of the matrix $U$ as stated by the authors. Please remember that EOF modes ($U_j \times C_j$), EOF basis functions $U_j$ and EOF coefficients $C_j$ are different.

Comment 2: (Page 4, lines 4 - 7) "The fluctuations ... dynamics". I disagree with this statement. Please refer to the above suggested works and explain correctly what the basis functions $U_j$, $j = 2, ..., 6$ are describing.

Comment 3: (Page 4, Figure 1 (a) and (b)) Please clarify in the text that the top left panels of Figure 1 (a) and Figure 1 (b) compare diurnal mean TEC with $U_1$ and solar flux index with $C_1$, respectively.

Comment 4: (Page 4, line 16) A period of 0 means a harmonic function of infinite angular frequency. The statement is incorrect.

Comment 5: (Page 4, Table 1) Please specify that the explained variances and cumulative variances are expressed in percentage.

Comment 6: (Page 5, lines 3 - 5) The sentence is wrong. The least square method is used to estimate EOF coefficients from the exact coefficients $C_j$ (and not GPS-derived TEC as mentioned) and model inputs. (Please see Equation 5).

Comment 7: Specify the inputs of the models in section 2.

Comment 8: (Page 7) Comment about the failure/inaccuracy of IRI in predicting TEC during storms. IRI and GPS satellites provide TEC up to 2000 km and 20,200 km altitude, respectively. Comment discuss about this and the plasmaspheric contributions.

Comment 10: (Page 6, Figure 3): Add Kp index as the authors have used it to select quiet days.

Comment 11: (Page 7, Figure 4): Specify in the text that top panels represents Dst index.

---

## Author Comment (AC2) · 11 Oct 2018

**Response to review comments #RC2**

**Major comments**

**Comment 1:** EOF analysis has been applied to TEC/foF2 modeling in low and middle latitudes, during geomagnetically quiet and storm conditions, and at regional/global scale (References below). In my point of view, the presented work seems to be a repetition of what has been done previously and its contribution towards research progress referring to existing works is not clear. I am suggesting the authors to revise the introduction and include the references provided above and then highlight briefly and concisely their contribution to research progress.

**Response:** It is true that similar work especially with the modeling technique has been done for other regions. A number of the references suggested have been included in section 3. The focus of this work is on regional trend estimation, and hence the contribution of this work is in terms of using the EOF model as a background in trend estimation and the nature of the trends over the African region. The need for such a study has been stated in the last paragraph of the introduction.

**Comment 2:** (Page 2, lines: 18 - 19). The authors mention that 2-hour GIM data was interpolated to 1-hour data and it is evident that during interpolation some errors are introduced. I am suggesting the authors to clarify how the interpolation method used in this study has been validated before being applied, and how errors due to interpolation will affect TEC modeling results.

**Response:** Given the different times for CODE's GIMs (odd hour before 2002, even hours from 2002-2014 and hourly after 2014) it was necessary to interpolate the data to provide a uniform sampling. Before the interpolation, we first extracted the VTEC for each longitude-latitude pair. The VTEC was then interpolated in time domain using linear interpolation. The choice for the linear function was because;

- Piece-wise linear functions are used for representation in the time domain while generating GIMs.

- Linearly interpolated CODE's GIMs have been compared with TOPEX/Jason TEC data (Jee *et al.*, 2010).

Since the basis vectors give the average daily trend over the entire period (1999-2017), we do not think that the choice of linear interpolation of the VTEC would substantially affect the model. In case of daily random errors due to the interpolation procedure, they will manifest in the higher order EOF modes, and these were discarded when modeling the TEC.

**Comment 3:** Discussion of the results and conclusions should be revised. The authors should highlight the main findings of the current work referring to previous works.

**Response:** Some of the major findings of this work include;

- The EOF-based TEC model provides a better background TEC over Malindi than the IRI

- Trend of TEC over MAL2 is positive.

- Confirmation of latitudinal variation in trends of TEC over African low latitudes and of negative trends dominating over the geomagnetic equator.

These have been clearly pointed out in the conclusion during the revision.

**Minor comments**

**Comment 1:** (Page 3, line 33) The six EOF modes are not elements of the matrix U as stated by the authors. Please remember that EOF modes ($U_j \times C_j$), EOF basis functions $U_j$ and EOF coefficients $C_j$ are different.

**Response:** Thank you for the observation. The sentence was rephrased to:
The basis vectors of the first six EOF modes in matrix U and their corresponding coefficients obtained using equation 3 are shown in Figure 1.

**Comment 2:** (Page 4, lines 4 - 7) "The fluctuations ... dynamics". I disagree with this statement. Please refer to the above suggested works and explain correctly what the basis functions U j , j = 2, ..., 6 are describing.

**Response:** Since the basis modes represent the contribution of each factor in influencing the variability in the data, their ordering may vary for the different data sources. For example, according to Dabbakuti and Ratnam (2017), the second and third order basis function represent the semidiurnal variation associated with the summer to winter annual variation and ionospheric anomaly feature due to prereversal enhancement respectively. However, Dabbakuti and Ratnam (2016) observed that the second and third order base functions describe the variability due to irregularities and scale disturbances. While for our data, the second basis function appears to be associated with prereversal enhancement. It is important to note that, the physical interpretation of the basis functions are normally difficulty due to their geometric nature (Hannachi *et al.*, 2007). To avoid subjective interpretation of the basis functions, we have deleted the sentence "The fluctuations observed in the higher order basis functions could be signatures of the different processes (such as traveling thermospheric disturbances (TADs)) that influence the low latitude plasma dynamics" since no statistical analysis was done for the higher order basis functions.

**Comment 3:** (Page 4, Figure 1 (a) and (b)) Please clarify in the text that the top left panels of Figure 1 (a) and Figure 1 (b) compare diurnal mean TEC with U1 and solar flux index with C 1 , respectively.

**Response:** Thank you for the suggestion. The following sentence was added in the text
The top left panels of Figure 1 (a) and Figure 1 (b) compare diurnal mean TEC with the first basis vector U1 and solar flux index with coefficients C1 of the first EOF mode respectively

**Comment 4:** (Page 4, line 16) A period of 0 means a harmonic function of infinite angular frequency. The statement is incorrect.

**Response:** It is true that the statement is incorrect. This was changed to:
The EOF coefficients were expressed as a sum of linear and harmonic functions following the

procedure of Zhang *et al.* (2009) as

**Comment 5:** (Page 4, Table 1) Please specify that the explained variances and cumulative variances are expressed in percentage.

**Response:** Percentage sign (%) has been added in Table 1

**Comment 6:** (Page 5, lines 3 - 5) The sentence is wrong. The least square method is used to estimate EOF coefficients from the exact coefficients C j (and not GPS-derived TEC as mentioned) and model inputs. (Please see Equation 5).

**Response:** Thank you for the observation. The sentence was corrected to:
The coefficients $a_{j1}$ to $f_{j3}$ in equations 4-7 were determined using a least squares fit to the EOF coefficients $C_j$ in equation 3

**Comment 7:** Specify the inputs of the models in section 2.

**Response:** We added the sentence below to specify the inputs to the model:
"Based on the observations in Table 2, it was reasonable to use F10.7av and Dst as inputs to model the solar and magnetic dependences respectively of TEC over Malindi. Since these parameters vary with the day of the year (DOY), our third input parameter was the DOY number".

**Comment 8:** (Page 7) Comment about the failure/inaccuracy of IRI in predicting TEC during storms. IRI and GPS satellites provide TEC up to 2000 km and 20,200 km altitude, respectively. Comment discuss about this and the plasmaspheric contributions.

**Response:** Thank you for the suggestions. It is expected that the altitude difference would result in lower IRI TEC than GPS TEC due to the plasmaspheric contribution to the TEC. However some observations (eg. Olwendo *et al.* 2012 show that IRI overestimates the GPS TEC during low solar activity years and during June solstice. Such a difference may not be accounted for in terms of the altitude

Like any other empirical model, the IRI is limited. The inaccuracy of IRI in predicting TEC during storms over Malindi probably leaves an open research question. There may be need to improve on the storm model used in IRI in order to capture the different storm time TEC responses.

**Comment 10:** (Page 6, Figure 3): Add Kp index as the authors have used it to select quiet days.

**Response:** The maximum Kp and the minimum Dst for each of the days has been included in the plots

**Comment 11:** (Page 7, Figure 4): Specify in the text that top panels represents Dst index.

**Response:** The following sentence was added in the text:
"The bottom and the top panels of Figure 4 show variation of the hourly diurnal TEC and Dst index respectively during selected major geomagnetic storms".

**References**

Jee G., Lee H. B., Kim Y. H., Chung J. K. and Cho J., 2010. Assessment of GPS global ionosphere maps (GIM) by comparison between CODE GIM and TOPEX/Jason TEC data: Ionospheric perspective. J. Geophys. Res.: space physics,115, p A10319

Dabbakuti J.R.K., Ratnam D. V., (2017). Modeling and analysis of GPS-TEC low latitude climatology during the 24th solar cycle using empirical orthogonal functions. Adv.Space. Res.

Dabbakuti J.R.K., Ratnam D. V., (2016). Characterization of ionospheric variability in TEC using EOF and wavelets over low-latitude GNSS stations. Adv.Space. Res 57, pp. 2427–2443

Hannachi A., Jolliffe I.T. and Stephenson D.B., 2007. Empirical orthogonal functions and related techniques in atmospheric science: A review. Int. J. Climatol., 27, pp. 1119–1152.

Olwendo O., Baki P., Cilliers P., Mito C. and Doherty P., 2012. Comparison of GPS TEC measurements with IRI-2007 TEC prediction over the Kenyan region during the descending phase of solar cycle 23. Adv. Space Res., 49, pp. 914–921.

Uwamahoro, J. C., and Habarulema J. B., 2015. Modelling total electron content during geomagnetic storm conditions using empirical orthogonal functions and neural networks. J. Geophys. Res. Space Physics, 120, pp. 11000–11012.

Zhang M.L., Liu C., Wan W., Liu L. and Ning B., 2009. A global model of the ionospheric F2 peak height based on EOF analysis. Ann.Geophys., 27, pp. 3203–3212.

---

## Author Response (AR1)

**Response to Reviewer#RC1**

**Comment 1**

Equation (1): Did it require to include a mean term?

**Response**

Thank you for the inquiry. A mean term is not required. Singular Value Decomposition factorizes a matrix  $M_{m \times n}$  into three other matrices U, S and V as given in Equation 2. To recover  $M_{m \times n}$ , we take the matrix product of U, S and V. The  $\sum_{j=1}^{n}$  in Equation 1 is for the sum of the product of the matrix entries in the multiplication of U by C, where C is given by Equation 2. Comment 2

**Page 3: Line 37: I cannot see the secondary maximum at about 20:00 UT from the Figure**

**Response**

Thank you for the observation. Critical analysis of Figure 1 (a) shows that there is no secondary maximum at about 20:00 UT as we had earlier stated. What appears at about 20:00 UT is an enhancement. The sentence was therefore rephrased to: Figure 1 (a) shows that the average diurnal TEC (red curve) over Malindi has a pre-dawn minimum at about 03:00 UT, a maximum at about 11:30 UT and an enhancement from 18:00 to 20:00 UT. The maximum at 11:30 UT is possibly due to increased ionization as the solar zenith angle is nearly zero over Malindi around this time. The post sunset increase in the TEC from 18:00 to 20:00 UT could be due to an enhancement in the eastward electric field before its westward reversal at night. This appears in page 4 lines 7-10

**Comment 3**

Figure 1: What is the red curve? No words are given.

**Response**

Thank you for the observation. What the red curve represents has now been clearly stated in the caption. The caption of Figure 1 was therefore changed to:

The first six basis functions (a) representing the diurnal variation and their coefficients (b) which show the long-term variation of TEC over MAL2. The red curves in the top left panels in (a) and (b) compare the diurnal mean TEC with the first basis vector U1 and the solar radio flux index measured at 10.7 cm wavelength (F10.7) with coefficients C1 of the first EOF mode respectively.

**Comment 4**

Page 4, Line 16: The statement is incorrect. Periods of 0 is not the linear variation.

**Response**

We acknowledge that a linear function is not of period zero. In fact, a linear function has infinite period. We have deleted the incorrect statement and then updated the sentence in page 5 lines 14-19 to page 6 lines 1-5 as follows.

We then expressed the EOF coefficients as a sum of linear and harmonic functions following

the procedure of Zhang et al. (2009) as

$$C_{j}(d) = B_{j1}(d) + B_{j2}(d) + B_{j3}(d)$$
(1)

The term  $B_{j1}(d)$  is to account for the linear variation of the EOF coefficients with solar and magnetic activities and is given by

$$B_{j1}(d) = a_{j1} + b_{j1}F10.7av(d) + c_{j1}Dst(d)$$
 (2)

The semiannual and annual variations in the EOF coefficients are represented in equation 1 by the harmonic terms  $B_{j2}(d)$  and  $B_{j3}(d)$  of periods half a year and 1 year (365.25 days) respectively expressed as

$$B_{j2}(d) = \left[a_{j2} + b_{j2}F10.7_{av}(d) + c_{j2}Dst(d)\right] \cos\left(\frac{2\pi d}{365.25}\right) + \left[d_{j2} + e_{j2}F10.7_{av}(d) + f_{j2}Dst(d)\right] \sin\left(\frac{2\pi d}{365.25}\right)$$
(3)

$$B_{j3}(d) = \left[a_{j3} + b_{j3}F10.7_{av}(d) + c_{j3}Dst(d)\right] \cos\left(\frac{4\pi d}{365.25}\right) + \left[d_{j3} + e_{j3}F10.7_{av}(d) + f_{j3}Dst(d)\right] \sin\left(\frac{4\pi d}{365.25}\right)$$
(4)

**Comment 5**

Equation (4): Why such presentation is reasonable or enough. No words to support it, no references are cited.

**Response**

We have provided in page 5 line 15 the reference by Zhang et al. (2009) in support of the method.

**Comment 6 a**

Equation (5): Same to Equation (4). Moreover, (1) F10.7 in Equation (5) and F10.7av in Equation (6)

**Response**

Thank you for the observation. In equation (5) it should have been F10.7av not F10.7. This was a typing error and has been corrected to:

$$B_{j1}(d) = a_{j1} + b_{j1}F10.7av(d) + c_{j1}Dst(d)$$

**Comment 6 b**

Is it enough to present the geomagnetic activity condition with Dst?

**Response**

(i) The Dst is based on measurements at near equatorial magnetic observatories (Honolulu, San Juan, Hermanus and Kakioka) and therefore it predominantly gives an indication of the equatorial ring current variations. Therefore Dst is more appropriate than AE since the observations are equatorial rather than auroral.

(ii) The Dst and Kp indices are not independent, so the best of these would be the one that gives the highest correlation with the TEC observations. Of the three magnetic indices considered, it is Dst that showed the highest correlation with the coefficients of the first EOF mode

|    | Solar indices |       |              | 1          | Magnetic indices |               |  |
|----|---------------|-------|--------------|------------|------------------|---------------|--|
|    | SSN           | F10.7 | $F10.7_{av}$ | Кр         | AE               | Dst           |  |
| C1 | 71.2          | 75.8  | 79.1         | 19.0(18.8) | 15.6(15.1)       | -23.1 (-23.0) |  |
| C2 | 36.6          | 35.4  | 37.2         | 9.5(8.7)   | 12.1 (10.0)      | -5.5(-6.0)    |  |
| C3 | 16.9          | 16.7  | 16.4         | 4.2(7.2)   | 4.6(9.1)         | -4.1(-1.6)    |  |
| C4 | 4.4           | 7.5   | 8.1          | 17.8(14.4) | 20.1(18.4)       | -13.3 (-4.0)  |  |
| C5 | 5.8           | 6.9   | 7.0          | 2.3(0.3)   | 1.3 (0.2)        | -6.8(-6.5)    |  |
| C6 | 0.2           | 1.9   | 1.8          | 1.8(0.7)   | 3.9(1.8)         | -4.4 (-3.1)   |  |

Table RC1: Correlation of the expansion coefficients with some of the commonly used solar and magnetic indices. In brackets are the coefficients at a lag of one day

which explained 96.8% of the TEC variance over MAL2.

**Comment 6 c**

Is the linear relation between TEC and Dst sufficient?

**Response**

Numerous studies have used linear terms of solar and magnetic indices (e.g. Lastovicka et al.(2017), Uwamahoro and Habarulema (2015), Ercha et al 2012, Zhang et al 2009) to model TEC. More so, our data set did not reveal either a second or higher order relation between the daily averaged TEC and Dst, but rather a weak linear relation. Based on the observations with our data set, we adopted the linear relation in our model.

**Comment 6 d**

Is there no time delay between the occurrence of a geomagnetic storm and its effect on the ionosphere?

**Response**

Scherlies and Fejer (1997) observed that the equatorial plasma responds to high latitude current disturbance in time scales of 1-12 hrs (short-term effect) and 20-30 hrs (long-term effect). Owing to the fact that we are using daily averages of Dst, the short term effect can not be considered in the modeling process. To investigate the long term effects, we determined the correlation between the EOF coefficients and the magnetic indices at zero and one day lags. Our correlation analysis (see Table RC1) did not reveal significant differences in the correlation coefficients at zero and one day lags. In fact the daily Dst values showed slightly higher correlation coefficients at zero lag with the EOF coefficients except for the second EOF mode. Based on these observations, we used the daily Dst values with zero lag to model the EOF coefficients. As mentioned earlier, such a representation has also been used by many authors (eg Lastovicka et al 2017, Uwamahoro and Habarulema (2015), Ercha et al 2012, Zhang et al 2009) to model TEC.

**Comment 6 e**

How to simulate/reproduce the solar effect of TEC is essential for successful model TEC and further estimate the trend of TEC. In this work, the solar index uses F10.7 av. It did not provide references (suggest to cite two key works, Richards et al., 1994: Richards, P. G., J. A. Fennelly, and D. G. Torr (1994), EUVAC: A solar EUV flux model for aeronomic calculations, J. Geophys Res., 99, 8981-8992; Liu et al., 2006: Liu, L., W. Wan, B. Ning, et al. (2006), Solar activity variations of the ionospheric peak electron density, J. Geophys. Res., 111, A08304,

doi:10.1029/2006JA011598).

**Response**

Thank you for the suggestions. Appropriate references provided in page 5 line 10 were cited as suggested

**Comment 6 f**

According to the investigation of Liu and Chen (2009) and Liu et al. (2009), the TEC is better to present as a second-order polynomial with F10.7, especially under the situation for estimating trend. References: Liu, L., and Y. Chen (2009), Statistical analysis on the solar activity variations of the TEC derived at JPL from global GPS observations, J. Geophys. Res., 114, A10311, doi:10.1029/2009JA014533. Liu, L., W. Wan, B. Ning, and M.-L. Zhang (2009), Climatology of the mean TEC derived from GPS Global Ionospheric Maps, J. Geophys. Res., 114, A06308, doi:10.1029/2009JA014244

**Response**

Thank you for the suggestion. We investigated the relationship between the daily mean TEC

Figure RC1: Scatter plots for (a) daily mean TEC and F10.7 (b) daily mean TEC and F10.7av

and F10.7av. As suggested by Liu et al. (2009), representing F10.7 as second order polynomial has smaller root mean square error (see Figure RC1). The difference in rmse in using a linear fit and that of second order polynomial appears to be small especially with F10.7av.

We further proceeded to model the TEC by representing F10.7av as second order polynomial using the equations below.

$$C_{j}(d) = B_{j1}(d) + B_{j2}(d) + B_{j3}(d)$$

where

$$B_{j1}(d) = a_{j1} + b_{j1}F10.7_{av}(d) + c_{j1}[F10.7_{av}(d)]^2 + d_{j1}Dst(d)$$

 $B_{j2}(d) = \left[a_{j2} + b_{j2}F10.7_{av}(d) + c_{j2}[F10.7_{av}(d)]^2 + d_{j2}Dst(d)\right] \cos\left(\frac{2\pi d}{365.25}\right) \\ + \left[e_{j2} + f_{j2}F10.7_{av}(d) + g_{j2}[F10.7_{av}(d)]^2 + h_{j2}Dst(d)\right] \sin\left(\frac{2\pi d}{365.25}\right) \\ + \left[e_{j2} + f_{j2}F10.7_{av}(d) + g_{j2}[F10.7_{av}(d)]^2 + h_{j2}Dst(d)\right] \sin\left(\frac{2\pi d}{365.25}\right) \\ + \left[e_{j2} + f_{j2}F10.7_{av}(d) + g_{j2}[F10.7_{av}(d)]^2 + h_{j2}Dst(d)\right] \sin\left(\frac{2\pi d}{365.25}\right) \\ + \left[e_{j2} + f_{j2}F10.7_{av}(d) + g_{j2}[F10.7_{av}(d)]^2 + h_{j2}Dst(d)\right] \sin\left(\frac{2\pi d}{365.25}\right) \\ + \left[e_{j2} + f_{j2}F10.7_{av}(d) + g_{j2}[F10.7_{av}(d)]^2 + h_{j2}Dst(d)\right] \sin\left(\frac{2\pi d}{365.25}\right) \\ + \left[e_{j2} + f_{j2}F10.7_{av}(d) + g_{j2}[F10.7_{av}(d)]^2 + h_{j2}Dst(d)\right] \sin\left(\frac{2\pi d}{365.25}\right) \\ + \left[e_{j2} + f_{j2}F10.7_{av}(d) + g_{j2}[F10.7_{av}(d)]^2 + h_{j2}Dst(d)\right] \sin\left(\frac{2\pi d}{365.25}\right) \\ + \left[e_{j2} + f_{j2}F10.7_{av}(d) + g_{j2}[F10.7_{av}(d)]^2 + h_{j2}Dst(d)\right] \sin\left(\frac{2\pi d}{365.25}\right) \\ + \left[e_{j2} + f_{j2}F10.7_{av}(d) + g_{j2}[F10.7_{av}(d)]^2 + h_{j2}Dst(d)\right] \sin\left(\frac{2\pi d}{365.25}\right) \\ + \left[e_{j2} + f_{j2}F10.7_{av}(d) + g_{j2}[F10.7_{av}(d)]^2 + h_{j2}Dst(d)\right] \sin\left(\frac{2\pi d}{365.25}\right) \\ + \left[e_{j2} + f_{j2}F10.7_{av}(d) + g_{j2}F10.7_{av}(d)\right] + \left[e_{j2} + f_{j2}F10.7_{av}(d) + g_{j2}F10.7_{av}(d)\right] + \left[e_{j2} + f_{j2}F10.7_{av}(d)\right] + \left[e_{j2} + f_{j2}F10.7_{$

 $B_{j3}(d) = \left[a_{j3} + b_{j3}F10.7_{av}(d) + c_{j3}[F10.7_{av}(d)]^2 + d_{j3}Dst(d)\right] \cos\left(\frac{4\pi d}{365.25}\right) \\ + \left[e_{j3} + f_{j3}F10.7_{av}(d) + g_{j3}[F10.7_{av}(d)]^2 + h_{j3}Dst(d)\right] \sin\left(\frac{4\pi d}{365.25}\right) \\ + \left[e_{j3} + f_{j3}F10.7_{av}(d) + g_{j3}[F10.7_{av}(d)]^2 + h_{j3}Dst(d)\right] \sin\left(\frac{4\pi d}{365.25}\right) \\ + \left[e_{j3} + f_{j3}F10.7_{av}(d) + g_{j3}[F10.7_{av}(d)]^2 + h_{j3}Dst(d)\right] \sin\left(\frac{4\pi d}{365.25}\right) \\ + \left[e_{j3} + f_{j3}F10.7_{av}(d) + g_{j3}[F10.7_{av}(d)]^2 + h_{j3}Dst(d)\right] \sin\left(\frac{4\pi d}{365.25}\right) \\ + \left[e_{j3} + f_{j3}F10.7_{av}(d) + g_{j3}[F10.7_{av}(d)]^2 + h_{j3}Dst(d)\right] \sin\left(\frac{4\pi d}{365.25}\right) \\ + \left[e_{j3} + f_{j3}F10.7_{av}(d) + g_{j3}[F10.7_{av}(d)]^2 + h_{j3}Dst(d)\right] \sin\left(\frac{4\pi d}{365.25}\right) \\ + \left[e_{j3} + f_{j3}F10.7_{av}(d) + g_{j3}[F10.7_{av}(d)]^2 + h_{j3}Dst(d)\right] \sin\left(\frac{4\pi d}{365.25}\right) \\ + \left[e_{j3} + f_{j3}F10.7_{av}(d) + g_{j3}[F10.7_{av}(d)]^2 + h_{j3}Dst(d)\right] \sin\left(\frac{4\pi d}{365.25}\right) \\ + \left[e_{j3} + f_{j3}F10.7_{av}(d) + g_{j3}[F10.7_{av}(d)]^2 + h_{j3}Dst(d)\right] \sin\left(\frac{4\pi d}{365.25}\right) \\ + \left[e_{j3} + f_{j3}F10.7_{av}(d) + g_{j3}[F10.7_{av}(d)]^2 + h_{j3}Dst(d)\right] \sin\left(\frac{4\pi d}{365.25}\right) \\ + \left[e_{j3} + f_{j3}F10.7_{av}(d) + g_{j3}F10.7_{av}(d)\right] + \left[e_{j3} + f_{j3}F10.7_{av}(d) + g_{j3}F10.7_{av}(d)\right] + \left[e_{j3} + f_{j3}F10.7_{av}(d)\right] + \left[e_{j3} + f_{j3}F10.7_{$

Figure RC2: Scatter plots for observed TEC and modeled TEC using (a) F10.7av as a linear term (b) F10.7av as a second order polynomial

Figure RC2 shows that representing the solar activity as a second order polynomial results in smaller error than using the linear relationship. However, the difference in the rmse appears insignificant and we think will not result in any substantial differences in the conclusions

**Comment 7**

Figure 2: An equation is welcome to give the regression. I am curious as to why there is a significant intercept.

**Response**

A regression equation has been included in Figure 2 (d) as shown in Figure RC3. A positive

---

## Referee Report (RR1)

The authors have considered all the comments suggested by other two reviewers.
However, I have some minor comments which I would like the authors to address properly before this work acceptance for publication in Annales Geophysicae.
But, first of all, the pdf version of the manuscript has some missing parts, such as indications and numbers in all the Figures. Also the axes title are missing. Please, check when you upload your revised version, that it looks exactly as your original file.

Comments:
1) You mention that form figure 1(b) you can deduce the annual and semi-annual variations. I do not see how you can deduce this. In the figures it can hardly be seen the annual variation.

2) Before section 3.2 you say "...from which the ionosphere owes its existence". I would delete this phrase since it is too strong. And, even if the Sun is the main player here, the ionosphere has many other forcings.

3) Correlation coefficients run from –1 to 1, so I guess that you have multipied them by 100 in Table 1. You should mention this in the Table, or divide your numbers by 100.

4) Regarding the intercept of 3.2 in Figure 2, maybe it is not significantly different from zero. So please, provide its error so we can check this.

5) Regardind the long-term trend value in Malindi, you should take into account that the magentic Equator has a secular displacement, with its consequencies in the trough and crests of the EIA. So surely, this may be another trend forcing at this location. You could mention this. It deserves another study which can be easily done for another work. I mean, to check how is the magnetic Equator shifting at the longitude of Malindi during the period here analyzed.
See for example Gnabahou, D. A., A. G. Elias, and F. Ouattara (2013), Long-term trend of foF2 at a West African equatorial station linked to greenhouse gas increase and dip equator secular displacement, J. Geophys. Res. Space Physics, 118, 3909–3913, doi:10.1002/jgra.50381.

6) It is not clear for me what does Figures 6(a) mean. 6(a)i is clear. I think that 6(a) ii, iii, and iv, show amplitude of variation for a certain periodicity. And figure 6(b) shows how this amplitude varies in time.
If this is so, you should explain more the spatial variability seen in 6(a) ii, iii and iv, which is not trivial. What does is mean the "counter phase" (blues in come regions and red in others) of the smaller scale of these figures?
Another possiblility is deleting this part from the paper.

7) Regarding section 5 dealing with long-term trends I see here the typical pattern of positive and negative trends in the map which could possibly due to the displacement of the magnetic equator there. Again, take a look at the paper by Gnabahou et al. (2013), to see if you can add a short discussion on this possibility.

---

## Author Response (AR2)

**Response to review comments #RC3**

**Comment 1:** You mention that from figure 1(b) you can deduce the annual and semi-annual variations. I do not see how you can deduce this. In the figures it can hardly be seen the annual variation.

**Response:** A Section of the Figure has been magnified and inserted in the top left panel of Figure 1 (b) to make the semi-annual variation clear. The caption of Figure 1 was therefore changed to:
The first six basis functions (a) representing the diurnal variation and their coefficients (b) which show the long-term variation of TEC over MAL2. The red curves in the top left panels in (a) and (b) compare the diurnal mean TEC with the first basis vector U1 and the solar radio flux index measured at 10.7 cm wavelength (F10.7) with coefficients C1 of the first EOF mode respectively. Inserted in the top left panel of (b) is a magnified section of the coefficients C1 for 2002-2003 to show the semiannual and annual variations.

**Comment 2:** Before section 3.2 you say "...from which the ionosphere owes its existence". I would delete this phrase since it is too strong. And, even if the Sun is the main player here, the ionosphere has many other forcings.

**Response:** Thank you for the observation. The words "from which the ionosphere owes its existence" were deleted. The sentence was therefore rephrased to:
These coefficients are well correlated with the solar radio flux measured at 10.7 cm wavelength (F10.7), confirming that the main driver of ionospheric variability over Malindi is the changes in the extreme ultraviolet (EUV) radiation. This appears in page 5 lines 4-6

**Comment 3:** Correlation coefficients run from –1 to 1, so I guess that you have multiplied them by 100 in Table 1. You should mention this in the Table, or divide your numbers by 100.

**Response:** It is true that the correlation coefficients have been multiplied by 100. A percentage sign was used in Table 1 to point out that the correlation coefficients are in percentages.

**Comment 4:** Regarding the intercept of 3.2 in Figure 2, may be it is not significantly different from zero. So please, provide its error so we can check this.

**Response:** Thank you for the suggestion. The error in the slope and the intercept have been included in the regression equation. The root mean square error has also been included in Figure 2(d).

**Comment 5:** Regarding the long-term trend value in Malindi, you should take into account that the magnetic Equator has a secular displacement, with its consequences in the trough and crests of the EIA. So surely, this may be another trend forcing at this location. You could mention this. It deserves another study which can be easily done for another work. I mean, to check how is the magnetic Equator shifting at the longitude of Malindi during the period here analyzed. See for example Gnabahou, D. A., A. G. Elias, and F. Ouattara (2013), Long-term trend of foF2 at a West African equatorial station linked to greenhouse gas increase and dip equator secular displacement, J. Geophys. Res. Space Physics, 118, 3909–3913, doi:10.1002/jgra.50381.

**Response:** Thank you for the suggestion. Since the relative contributions of the different trend driving mechanisms at Malindi have not been investigated in this work, we preferred to mention secular variation of the dip angle as a possible driver in the general discussion together with the suggestion in comment 7. This mention appears in page 14 lines 2-4 as

While the geomagnetic control of the trends is either due to the long-term changes in geomagnetic activity (Danilov and Mikhailov, 1999; Mikhailov and Marin, 2000) or through Earth's magnetic field secular variations (Foppiano *et al.*, 1999; Gnabahou *et al.*, 2013). This latter view may explain the latitudinal dependence in the trends over African low latitude region

**Comment 6:** It is not clear for me what does Figures 6(a) mean. 6(a)i is clear. I think that 6(a) ii, iii, and iv, show amplitude of variation for a certain periodicity. And figure 6(b) shows how this amplitude varies in time. If this is so, you should explain more the spatial variability seen in 6(a) ii, iii and iv, which is not trivial. What does is mean the "counter phase" (blues in come regions and red in others) of the smaller scale of these figures? Another possibility is deleting this part from the paper.

**Response:** Thank you for the suggestions. Figure 6(a) represent the first four basis modes that explain most of the variation in the data. Figure 6(b) indeed show how the amplitudes (coefficients) of these basis modes change with time. We compared Figure 6(a)(i) with the average TEC (not shown) and we concluded that the first mode is related to the average diurnal TEC. We used the periodicity (alternate blue and red regions) in Figure 6(b)(ii) that possibly represent the equinoxes and the solstice to arrive at a conclusion that the second basis mode in Figure 6(a)(ii) is related to the different seasons. This conclusion is similar to what Ercha *et al.* (2012) observed. The spatial variation seen in 6(a)(iii), clearly suggests that the third EOF basis mode is associated with a phenomena related to the equatorial ionization anomaly. The fourth mode in Figure 6(a)(iv) is quite difficult to identify due to the geometric nature of the EOF modes (Hannachi *et al.*, 2007). We included it in Figure 6 just for completeness. However, since all the above analysis are not much connected to the modeling procedure, we have deleted Figure 6 and all related explanations.

**Comment 7:** Regarding section 5 dealing with long-term trends I see here the typical pattern of positive and negative trends in the map which could possibly due to the displacement of the magnetic equator there. Again, take a look at the paper by Gnabahou et al. (2013), to see if you can add a short discussion on this possibility.

**Response:** Thank you for the suggestion. We have added a short discussion in relation to the geomagnetic factor as driver of trends as

"While the geomagnetic control of the trends is either due to the long-term changes in geomagnetic activity (Danilov and Mikhailov, 1999; Mikhailov and Marin, 2000) or through Earth's magnetic field secular variations (Foppiano *et al.*, 1999; Gnabahou *et al.*, 2013). This latter view may explain the latitudinal dependence in the trends over African low latitude region". This appears in page 14 lines 2-4

**References**

Danilov A. D. and Mikhailov A. V, 1999. Spatial and seasonal variations of the foF2 long-term trends. Ann. Geophys., 17, pp 1239-1243

Ercha A., Zhang D., Ridley A. J., Xiao Z., and Hao Y., 2012. A global model: Empirical orthogonal function analysis of total electron content 1999–2009 data. J. Geophys. Res., 117, p A03328

Foppiano A.J., Cid L and Jara V, 1999. Ionospheric long-term trends for South American mid-latitudes. J. Atmos. Sol.-Terr., 61, pp 717–723

Gnabahou A. D, Elias G. A and Ouattara F, 2013. Long-term trend of f o F 2 at a West African equatorial station linked to greenhouse gas increase and dip equator secular displacement. J. Geophys. Res.: space physics,118, pp 3909–3913

Hannachi A, Jolliffe I. T and Stephenson D. B, 2007. Empirical orthogonal functions and related techniques in atmospheric science: A review. Int. J. Climatol., 27, pp 1119–1152

Mikhailov A. V. and Marin D., 2000. Geomagnetic control of the foF2 long-term trends. Ann. Geophys., 18, pp 653–665